# The macroevolutionary impact of recent and imminent mammal extinctions on Madagascar

Nathan M. Michielsen [1,2], Steven M. Goodman [3,4], Voahangy Soarimalala[4], Alexandra A. E. van der Geer [1], Liliana M. Dávalos [5,6], Grace I. Saville [1], Nathan Upham [7] & Luis Valente [1,8] ✉

Many of Madagascar's unique species are threatened with extinction. However, the severity of recent and potential extinctions in a global evolutionary context is unquantified. Here, we compile a phylogenetic dataset for the complete non-marine mammalian biota of Madagascar and estimate natural rates of extinction, colonization, and speciation. We measure how long it would take to restore Madagascar's mammalian biodiversity under these rates, the "evolutionary return time" (ERT). At the time of human arrival there were approximately 250 species of mammals on Madagascar, resulting from 33 colonisation events (28 by bats), but at least 30 of these species have gone extinct since then. We show that the loss of currently threatened species would have a much deeper long-term impact than all the extinctions since human arrival. A return from current to pre-human diversity would take 1.6 million years (Myr) for bats, and 2.9 Myr for non-volant mammals. However, if species currently classified as threatened go extinct, the ERT rises to 2.9 Myr for bats and 23 Myr for non-volant mammals. Our results suggest that an extinction wave with deep evolutionary impact is imminent on Madagascar unless immediate conservation actions are taken.

The island of Madagascar is renowned for its exceptional biodiversity and levels of endemism at different taxonomic levels, which evolved over millions of years in isolation[1,2]. Like most islands, Madagascar underwent substantial levels of extinction, predominantly of large-bodied animals, coinciding with the period since human arrival and population expansion[3,4]. Unlike many other tropical islands, however, Madagascar still retains a large proportion of its native flora and fauna, probably due to a delayed increase in anthropogenic pressures following human arrival in combination with its large surface area of nearly 590,000 km², which is approaching continental regions in size[5,6]. Nevertheless, its extant biota faces important conservation challenges[7], with over 3500 Malagasy species of plants and animals considered in the Red List of the International Union for Conservation (IUCN) as being under threat (41% of total species from the island listed)[8]. The main anthropogenic pressures include land use conversion for agriculture, other forms of habitat degradation, invasive species, climate change and hunting[7,9,10]. Due to Madagascar's disproportionate contribution towards global biodiversity and endemism in relation to its surface area[11,12], and its status as one of the world's biodiversity hotspots[13], the island is a crucial system on which to measure human impact on biodiversity.

[1]Naturalis Biodiversity Center, Leiden, The Netherlands. [2]Institute for Biodiversity and Ecosystem Dynamics, University of Amsterdam, Amsterdam, The Netherlands. [3]Field Museum of Natural History, 1400 South DuSable Lake Shore Drive, Chicago, IL 60605, USA. [4]Association Vahatra, BP 3972 Antananarivo 101, Madagascar. [5]Department of Ecology and Evolution, Stony Brook University, Stony Brook, NY 11794, USA. [6]Consortium for Inter-Disciplinary Environmental Research, Stony Brook University, Stony Brook, NY 11794, USA. [7]School of Life Sciences, Arizona State University, Tempe, AZ 85287, USA. [8]Groningen Institute for Evolutionary Life Sciences, University of Groningen, Groningen, The Netherlands. ✉e-mail: luis.valente@naturalis.nl

In this study, we investigate to what extent humans have perturbed Madagascar away from its natural pre-human state, and what future perturbation we may expect in the Anthropocene. We do this by measuring how long it would take to restore the island's lost and threatened biodiversity. Counting the number of lost and threatened species provides a good quantification of the severity of an extinction episode (human caused or natural)[14]. However, species diversity evolves at different rates around the globe[15–17], so that two islands may have lost the same number of species, but if average natural local rates of colonization and speciation are lower for one of the islands[18], it will take longer for that island to "recover" the lost diversity in an evolutionary context. Geography, regional environmental factors and the biological traits of a taxon each influence the rates at which new species colonize, evolve and go extinct. As a result, these rates vary across geographical regions[18,19] and taxonomic groups[20]. Thus, an alternative approach offering additional insight on human impact is to measure the evolutionary return time (ERT), that is to say the time it takes for the number of species in a region to return to a given species diversity level[21,22]. The ERT differs from other approaches that measure loss of evolutionary history (e.g. refs. 23, 24), by explicitly considering regional rates of species assembly (colonization, speciation and natural extinction) when assessing the impact of extinctions. While other methods focus on recovering phylogenetic diversity (measured by branch lengths, for example across the entire mammalian tree[24]), the ERT focuses on the recovery of taxonomic diversity (measured in number of species) via local evolutionary and biogeographical processes. In this study our focus is therefore on the (temporal) recovery of taxonomic diversity, by which we mean the number of species present on Madagascar.

Estimating ERT requires knowledge of the number and causes of recent extinctions, as well as of phylogenetic relationship between species[21]. However, such knowledge is currently unavailable for most groups of Malagasy organisms. One of the few exceptions are mammals, for which decades of taxonomic, palaeontological, molecular and conservation work exists[4,25–27]. Palaeontological research indicates Holocene extinctions of dozens of Malagasy mammal species, which fit into a wider pattern of relatively rapid extinctions of large body-sized animals that took place after human arrival[4,6,28,29]. The causes of recent mammalian extinctions on islands, i.e. those that have taken place in the late Holocene in a period largely coinciding with human settlement, are however, disputed. First, there is uncertainty regarding the times of extinction on Madagascar. For example, for the bats *Macronycteris besoaka* and *Paratriaenops goodmani*, the fossil record is too incomplete to confidently infer whether extinction occurred before or after initial human arrival[30]. Second, there is debate regarding the date humans first established on the island. The earliest evidence of human activity dates back to ~10,500 years before present (years BP) ([31], but see[5,32]). Then, evidence of human presence largely disappears for ~8000 years, with the exception of a few intermediate dates for which human presence can be inferred[31,33]. Thus we consider the period of established and continuous human habitation to be the time frame encompassing ~2500 years BP onward[3–5,32,33], when the human population expanded and started to have a strong impact on the island's ecosystems. Third, there is uncertainty about whether the recent extinctions were mainly caused by humans, natural changes or a combination of both[6,34–37]. For some species, an anthropogenic cause of extinction has been suggested[38]. For others, a natural cause of extinction is favoured, for instance, the rodent *Nesomys narindaensis*, thought to have gone extinct as its environment became drier due to natural climate change[39]. For other species, such as the extant Malagasy giant jumping rat (*Hypogeomys antimena*), which in the Quaternary had a broad distribution, it is difficult to disentangle human and natural causes for range reduction, and perhaps a combination of both may be the best explanation[40]. The same conclusion has been reached based on different types of information and datasets[6,34].

The mammalian fauna that did survive until the present is still taxonomically diverse, although the largest bodied species are extinct. However, the extant fauna is highly endangered, with over half of the island's mammal species currently classified as threatened with extinction by the IUCN[8]. The number of species considered threatened has increased substantially in the last decades[8], because: (1) the intensification of human threats on the island[10] has led many species to move from non-threat to threat categories, (2) renewed biological inventories on the island are bringing new information on species distributions, (3) many new species have been discovered or described and (4) a large percentage of species have only recently been evaluated by the IUCN. Whether this recent increase in threatened species would have a disproportionate impact on estimates of ERT if these species eventually go extinct will depend on the extent to which the number of species lost is a good surrogate for ERT.

Here, we capitalize on the extensive research over the past decades conducted on different aspects of Malagasy mammals to build a comprehensive new dataset describing the pattern of accumulation of species (via colonization, speciation and extinction) for this speciose insular mammal fauna. We then estimate the ERT for Malagasy mammals under a series of scenarios accounting for uncertainty in evolutionary rates and the times and causes of extinction. We also assess how recent changes in threat status affect the estimates of evolutionary assembly history that Madagascar stands to lose.

## Results
### Malagasy mammalian diversity
We compiled a checklist of all non-marine mammalian species known to have been present on Madagascar in the late Holocene prior to human arrival and population expansion ~2500 years ago (Supplementary Data S1). The checklist includes all extant native species and all those hypothesized to have gone extinct in the late Holocene. For the extinct species, we compiled information on the timing and causes of extinctions, and for the extant species we obtained their IUCN threat status in the 2010, 2015 and 2021[8,41,42] Red List assessments. In our checklist we identify a total of 249 species of mammals that were, based on different lines of evidence, present on Madagascar at the time of human arrival. Of these, 46 are bats and 203 are non-volant. All species are endemic, except for nine bat species. The largest mammalian clade comprises the lemurs (Lemuroidea) with 126 species at the time of human arrival, 17 of which have since gone extinct. A total of 30 of the 249 species have gone extinct recently (approximately the last 2500 years), of which 28 were non-volant and two were bats. Of the 30 recent extinctions, we found that 16 have been proposed to have an anthropogenic link, four have a relatively well-established natural cause and for 10 the cause is uncertain (column "Extinction cause" in Supplementary Data S1). For nine species, it is unclear whether extinction pre-dates or post-dates human arrival (column "Extinction Before/After Humans" Supplementary Data S1).

The number of extant species classified as threatened (species classified as Vulnerable, Endangered or Critically Endangered) by the IUCN has increased substantially over the last 10 years. There were 56 species classified as threatened in the IUCN Red List of 2010, 110 in 2015 and 128 in 2021 (Supplementary Data S1). A large proportion of the increase in threatened species accrues from increased knowledge: 57 of the species that are currently threatened had not been evaluated or were Data Deficient in 2010. However, 15 species were evaluated in 2010 and have changed from a non-threat to a threat category in 2021 (representing 21% of the changes). One hundred four of the extant lemurs (95%) are threatened with extinction, and of these 33 are classified as Critically Endangered (30% of all extant lemurs) (Fig. 1). A total of 6 tenrecs, 6 euplerid carnivorans, 5 bats and 7 nesomyine rodents are currently threatened with extinction (Fig. 1).

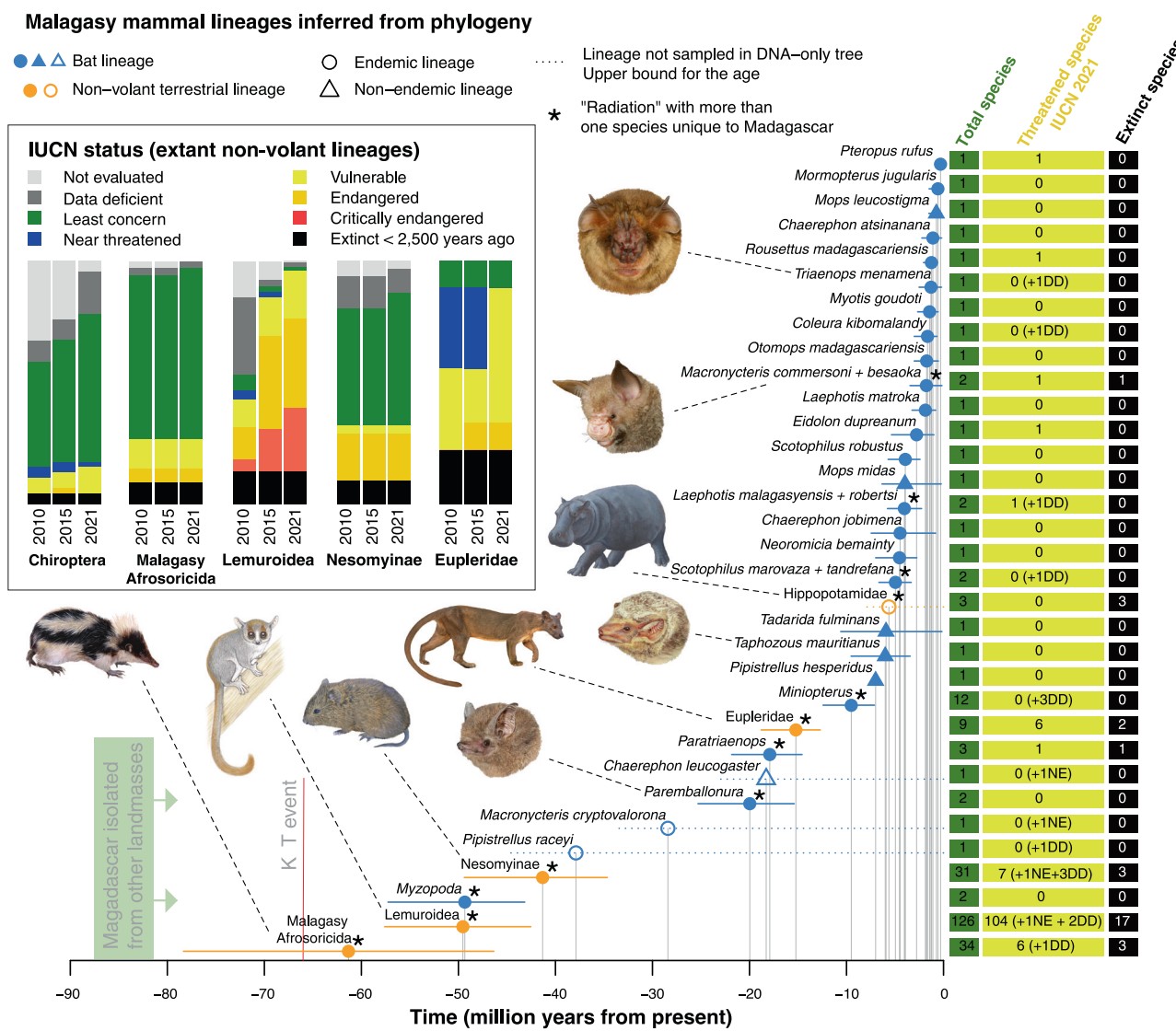

**Fig. 1 | Mammalian lineages present on Madagascar at the time of human arrival.** Colonization times in millions of years (Myr) of all Malagasy non-marine mammalian lineages descending from separate colonization events inferred from the phylogenetic data, assuming colonization scenario 1. Based on 1000 dated trees from the posterior distribution of the mammal trees derived from[43]. Horizontal lines indicate 95% highest posterior density for the colonization times; filled symbols on top of the lines indicate the mean of the distribution (*n* = 1000). For cases for which the lineage was not sampled in the phylogenetic trees (dashed horizontal lines), only a maximum age for the colonization event can be inferred (see Methods). For such cases, the symbols on top of the lines are given for visibility purposes

only. The barplots show the IUCN status of species in the extant non-volant clades. Images of species show a selection of representatives of different lineages: Afrosoricida (*Hemicentetes nigriceps*), Lemuroidea (*Microcebus griseorufus*), Nesomyinae (*Brachyuromys betsileoensis*), Eupleridae (*Cryptoprocta ferox*), Emballonuridae (*Paremballonura tiavato, Taphozous mauritianus*), Hippopotamidae (*Hippopotamus* sp.), Hipposideridae (*Macronycteris commersoni*) and Rhinonycteridae (*Triaenops menamena*). Lemur illustration copyright 2013 Stephen D. Nash / IUCN SSC Primate Specialist Group. *Hippopotamus* by Alexis Vlachos (courtesy George Lyras). All other illustrations copyright Velizar Simeonovski. Used with permission.

## Phylogenetic information

We extracted phylogenetic information on the number of colonization events, the estimated dates of ancestral colonization of Madagascar, the number of species within each monophyletic lineage and the timing of within-island speciation events from a recent comprehensive dated phylogenetic analysis of mammals, which includes 5911 species[43]. The number of colonization events of Malagasy mammals inferred from the phylogenetic data varies because the placement of some taxa differs across the posterior distribution of trees. We considered two alternative colonization scenarios (CS), one in which we favour fewer colonizations (CS1) and one in which we favour more (CS2) (Table S1, Supplementary Data S2, S3). Under both scenarios, the number of bat colonizations greatly exceeds that of non-volant mammals. Under CS1, we identified 33 lineages of mammals, of

which 28 were bats, present at the time of human arrival and resulting from independent colonizations of Madagascar (Fig. 1). Under CS2 there were 39 colonizations, 32 by bats.

The earliest colonization of an extant mammal group present at the time of human arrival are the tenrecs (CS1), which arrived on Madagascar ~61.3 (46.4–78.3, 95% highest posterior density interval, HPD) Myr ago, followed by the lemurs 49.5 (42.5–57.6) Myr ago (Fig. 1, Supplementary Data S2). Under CS2, assuming an independent colonization of Bibymalagasia (*Plesiorycteropus*), this lineage may have colonized Madagascar up to a maximum of 78.0 (62.3–88.0) Myr ago (Supplementary Data S3). The oldest bat clade on Madagascar is the endemic family Myzopodidae represented by the genus *Myzopoda*, whose ancestor colonized Madagascar 49.3 (43.2–57.3) Myr ago. The most recent mammalian colonist of Madagascar is the Madagascar

flying fox, *Pteropus rufus*, which arrived 0.31 Myr ago (0.08–0.65). These estimates, based exclusively on the phylogeny by ref. 43, are in agreement with previous phylogenetic studies[2,44–46].

## Macroevolutionary models

We estimated the average natural rates of colonization, extinction and speciation (CES rates), i.e. the biogeographical and diversification rates at which the Malagasy mammalian community assembled in the absence of humans, by fitting DAISIE[18,47] to the phylogenetic data. DAISIE is an island biogeography model that allows estimating island-specific CES rates based on the number of species, colonization and branching times and endemicity status of a given insular community[47]. Instead of focusing on a single radiation, DAISIE considers all clades of the target community resulting from independent colonizations. The method can estimate rates of colonization (how often species colonize the island), speciation via cladogenesis (speciation with in situ lineage splitting), anagenesis (speciation without lineage splitting) and natural extinction (how often species go extinct from the island under natural rates, i.e. the rates before human arrival). DAISIE can also estimate a carrying capacity (*K*), the maximum number of species a given clade can attain on the island, which will be infinite when there is no diversity dependence in rates, or finite when there is diversity dependence in rates of colonization and cladogenesis.

To account for uncertainty in various aspects of the data we produced a series of datasets (D1–D13, Supplementary Data S4) differing in the number of species in the mainland pool thought to be able to colonize Madagascar, the island age, the number of colonization events of the island inferred from the phylogeny, the level of human impact (more or fewer species assumed to have an anthropogenic extinction cause) and the completeness of the phylogenetic dataset. We fitted a set of 30 DAISIE models (Table S2) to all datasets. Models M1–M4 assume homogeneous CES rates for all mammals on Madagascar, while for models M5–M30 we allow one or more of the CES parameters to vary between non-volant mammals and bats. Full results for all datasets D1–D13 are shown in Supplementary Data S4, but as results did not vary strongly between datasets, we focus in the main text on the "main dataset" (D1, mainland pool of 1000 species, island age of 88 Myr, fewer colonization events favoured, high human impact and using trees with only species that were included using DNA sequence data in the phylogeny[43]). We also fitted two time-variable shift models[48] (M31 and M32) to dataset D1, in which we allowed for colonization rates to change at a certain point in time (Table S3). The preferred model for the main dataset is M26 (Table S3). Under M26 (time-constant model), bats have a higher rate of extinction and a much higher rate of colonization than non-volant mammals, but otherwise share the same speciation rates (cladogenesis and anagenesis). Under this model, non-volant mammals have non-equilibrium dynamics—their diversity increases steadily through time with no upper bound. In contrast, bats follow an equilibrium model, under which diversity tends towards a constant value of ~57 species, as determined by a higher rate of extinction than of cladogenesis. This is a sink-equilibrium scenario in which the island needs to receive new colonist species from the mainland to maintain its diversity, otherwise all species would eventually disappear from the island, which resembles the scenario previously proposed for noctilionoid bats of the Greater Antilles[22]. The sink equilibrium detected in bats differs from a diversity-dependent equilibrium model. Under diversity dependence there is an upper bound to species diversity on the island driven by competition for limited niches and set by a diversity carrying capacity (*K*), but this was rejected for bats. In the sink scenario, the equilibrium is not driven by diversity dependence in rates, but instead controlled by the high rate of extinction.

The M26 model was preferred for all trees from the posterior (parameters in Table S4). Simulations of the M26 model revealed a good fit to the data (Fig. S1).

## Evolutionary return time (ERT)

The ERT can be seen as the time it takes for the insular community to return to a certain number of species under natural conditions (without anthropogenic interference) (See Methods). To estimate ERT, we counted the number of species that were present on Madagascar in the late Holocene prior to human arrival (pre-human diversity) (Table S5). We consider the pre-human diversity to be the island's natural diversity and ideal conservation target (as tends to happen in restoration projects), as this is the diversity that arose through millions of years of ecological, evolutionary and biogeographical processes without human impact (it can be argued that anthropogenic extinctions are also natural, but these are generally much higher than background rates). We also counted the number of mammal species that will remain extant on the island if currently threatened species go extinct. We classified species as threatened if they were assigned one of the following IUCN categories: Vulnerable (VU), Endangered (EN) or Critically Endangered (CR) (Fig. 2). The diversity of Malagasy mammals at different stages—pre-human diversity, contemporary diversity or diversity if currently threatened species go extinct—are shown in Table S5. In the pessimistic scenario that all currently threatened species will go extinct (IUCN 2021), only 91 species of mammals out of a total of 219 (current diversity) would survive on the island.

We estimated the ERT of Malagasy mammals for: (1) the return from current diversity to pre-human diversity, and (2) the return from the diversity that will remain if threatened species (VU + EN + CR) go extinct back to current diversity. Note that pre-human and contemporary diversities differ between non-volant species and bats (Figs. 3, S3 and Table S5), and that once these respective target diversities have been reached, diversity can continue to increase (up to the equilibrium diversity in case there is one, as is the case in bats). To calculate ERT we simulated species diversity into the future using DAISIE with the parameters of the best model for each dataset (D1–D13). The results for all sensitivity datasets (D1–D13) are shown in Table S6 and the Supplementary Methods. Differences were mostly subtle, so here we discuss only the estimates for the main dataset (D1).

For D1, for non-volant mammals, the ERT from contemporary to pre-human diversity is 2.9 (2.3–3.6; 2.5–97.5 percentiles) Myr (Figs. 2, 3 and S2, Table S7). For non-volant mammals the ERT if threatened species go extinct has increased from 7.1 (5.6–8.7) Myr (2010), to 18 (14.2–22) Myr (2015) to 23.3 (18.5–28.3) Myr (2021) (Figs. 2, 3, S2, S3, Table S7). In other words, the ERT increased by ~16 Myr between 2010 and 2021 associated with threats to the island's mammal fauna. The ERTs for bats are generally lower, given there are fewer extinct and threatened species based on IUCN assessments (Figs. 2, 3, S2, S3, Table S5, S7). For bats, the ERT from contemporary to pre-human diversity is 1.6 (1.2–2.2) Myr (Figs. 2, 3 and S2, Table S7). The ERT for bats if threatened species go extinct has increased from 1.9 (1.5–2.4) Myr (2010), to 2.4 (1.9–3) Myr (2015) to 2.9 (2.3–3.6) Myr (2021) (Figs. 2, 3, S2, S3, Table S7). If all non-evaluated species that remain to be assessed following IUCN Red List criteria were evaluated as threatened, the ERT to return to contemporary diversity would rise to 26.2 (20.8–32) Myr for non-volant mammals (~13% increase) and 6.6 (5.5–7.8) Myr for bats (more than double) (Table S8, scenario A). If only species classified as CR in 2021 go extinct, the ERT is much shorter (Table S8, scenario B).

If we consider only species that have changed from non-threat to a threat category between 2010 and 2021, and for which there has been no change in species taxonomy between those assessments (10 species, 9 non-volant and one bat, Supplementary Data S5), the ERT for non-volant mammals would be 8.7 (6.9–10.7) Myr, an increase of ~2 Myr between 2010 and 2021, and for bats 2.42 (1.92–3.02) Myr, an increase of ~0.5 Myr (Table S8, scenario C).

The ERTs we estimate focus on the local Malagasy diversity, not on global diversity. Global diversity will only increase when a new endemic species evolves on Madagascar. For example, colonization of

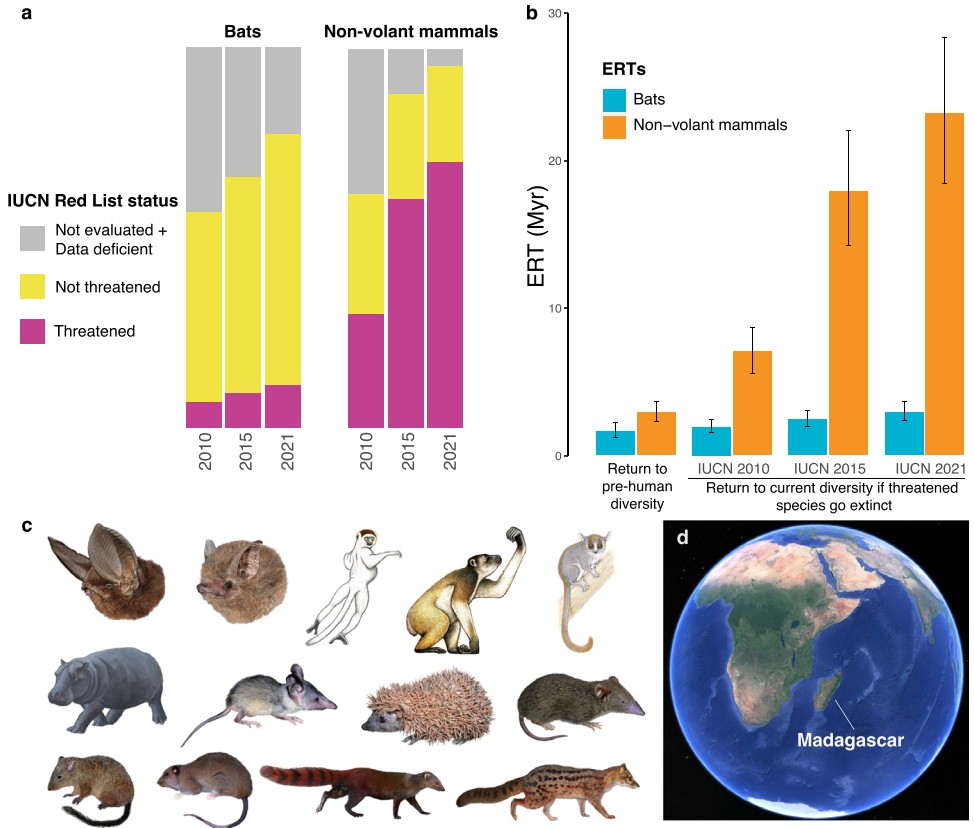

**Fig. 2 | IUCN status of Malagasy mammals and the associated ERTs. a** Change in IUCN Red List status between 2010 and 2021 for species of bats and non-volant mammals. IUCN Categories: Not threatened: includes Least Concern and Near Threatened; Threatened: includes Vulnerable, Endangered and Critically Endangered. **b** The evolutionary return times (ERTs) in millions of years (Myr) estimated for different scenarios of human impact for the main dataset (D1) under the M26 model. Barplots show the mean values and the error bars are the 2.5–97.5 percentiles across the posterior distribution of trees, $n = 1000$ (full distribution shown in Fig. 3). **c** Examples of extant and extinct Malagasy mammals. From left to right,

top to bottom: Chiroptera (*Myzopoda aurita*, *Paremballonura tiavato*); Lemuroidea (*Propithecus verreauxi*, *Archaeoindris fontoynontii* (extinct), *Microcebus griseorufus*), Hippopotamidae (*Hippopotamus* sp. (extinct)); Afrosoricida (*Geogale aurita*, *Echinops telfairi*, *Microgale brevicaudata*); Nesomyinae (*Nesomys lambertoni*, *Brachytarsomys villosa*); Eupleridae (*Galidia elegans*, *Fossa fossana*). **d** Contemporary configuration of Madagascar. Lemur illustrations copyright 2013 Stephen D. Nash / IUCN SSC Primate Specialist Group. *Hippopotamus* by Alexis Vlachos (courtesy George Lyras). All other illustrations copyright Velizar Simeonovski. Used with permission. Map data copyright 2022 Google.

Madagascar by a mainland species can lead to higher diversity values on the island if the species is not already present there, but global diversity would not change. This scenario will be particularly common when colonization rates are high and speciation rates are low. However, in the estimated parameters of our model for non-volant mammals, the rates of speciation (both anagenesis and cladogenesis) are much higher than rates of colonization, and thus a return to a target global diversity (e.g. return to pre-human number of species endemic to Madagascar) is achieved in a similar time frame as a target local diversity (Table S9). In the case of bats, a target global diversity is achieved more rapidly than the target local diversity, because both colonization and speciation rates are high and the number of endemic species to recover is low (Table S9).

**Species diversity lost versus ERT**

We ran simulations to measure how ERT varies with the number of extinct species, which revealed that the number of species lost is not a good surrogate of ERT (Fig. 4). For non-volant mammals, ERT increases steeply with the number of species lost when diversity is low (Fig. 4, yellow line, return to half of contemporary diversity), but at later stages, when diversity is higher, it increases less steeply (Fig. 4, blue and black lines). This is because the parameters of the M26 model for non-volant species are not in equilibrium, and diversity initially increases slowly and therefore ERTs become longer, as it takes longer to recover species. Conversely, at later stages of the diversity curve

(when diversity is higher), diversity accumulates rapidly, and therefore it becomes faster to recover species, and ERTs become shorter. For bats, the parameters of the M26 model are in equilibrium, which means that initially diversity increases very rapidly, and therefore ERTs are low (Fig. 4, yellow line). At later stages, when diversity is close to equilibrium and reaches a plateau, ERTs become longer (Fig. 4, blue and black lines). In other words, for non-volant mammals, for the same number of extinct species that need to be recovered, ERTs are lower at later stages of the island diversity curve; but for bats, ERTs are higher at later stages of the curve.

Previously undescribed extinct and extant species of mammals are likely to be discovered on Madagascar in the near future, and taxonomic revisions may lead to species splits, resulting in additional threatened species. We therefore also estimated the impact this may have in our ERT calculations (see Supplementary Methods). Assuming 30 new species (extinct, extant or both) are discovered in the next 10 years, we found that the ERT would generally rise substantially for bats (as it takes longer to recover bat species), but only moderately for non-volant mammals (Fig. S4). Under certain scenarios, the ERT would actually decline if more species are discovered, because the estimated rates of diversification and colonization would also rise (Fig. S4).

## Discussion

Here, we have built an unprecedented dataset containing taxonomic, phylogenetic, palaeontological, extinction and threat status

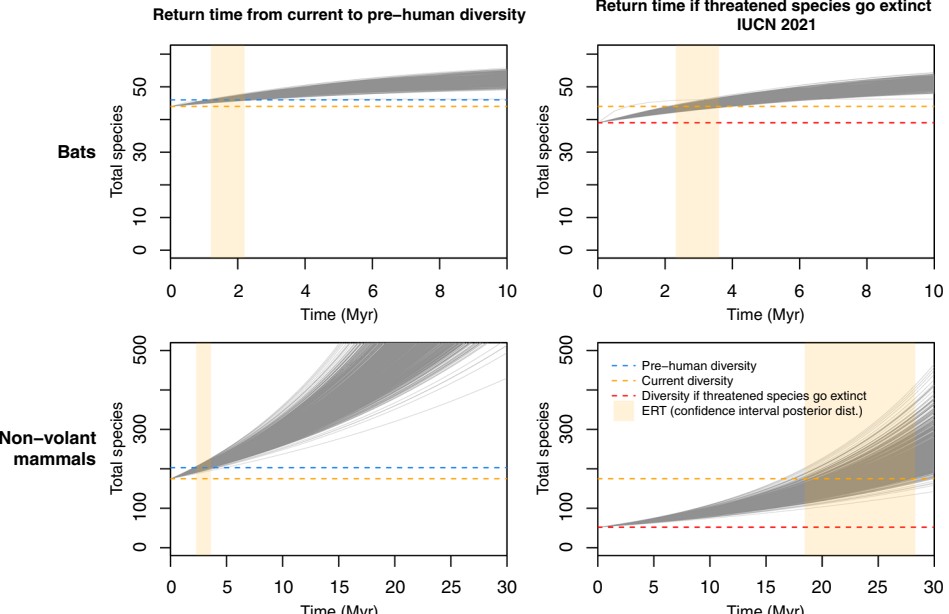

**Fig. 3 | Expected future diversity for bats and non-volant mammals on Madagascar.** Two scenarios are shown: return time from current to pre-human diversity (left panels); and return time to contemporary diversity if species classified as threatened by the IUCN in 2021 go extinct (right panels). Based on fitting the M26 model to 1000 trees from the posterior distribution of the main dataset, D1. The evolutionary return time (ERT) for each tree is the time it takes to go from the start diversity to the target diversity (e.g. red horizontal line to orange horizontal line in the bottom right plot). The vertical shaded area shows the 2.5–97.5 percentile of the ERT values based on the posterior distribution of trees.

information for the known Malagasy mammalian fauna, consisting of 249 species. We used this dataset to measure the ERT for this unique fauna to gain insight into the severity of Anthropocene extinctions on this island approaching continental scale. It would take 1.6 (1.2–2.2) Myr to recover the number of species of bats that has been lost since humans started having a substantial impact on Malagasy ecosystems ~2500 years ago and 2.9 (2.3–3.6) Myr to recover the number of non-volant terrestrial species. One way to interpret these figures is to consider that all those millions of years of evolutionary history have been nullified with the extinction of their end products in the last 2500 years, mostly by human activity. Even in a scenario in which human impact has been minimal, we found that the ERT is still over 1.7 Myr for non-volant mammals (Table S6; under the low human impact scenario no bat species is known to have gone extinct from human causes).

While of considerable importance, the impact of the initial episode of extinctions (30 species) on Madagascar in terms of ERT is lower than that previously estimated for the bat fauna of the Greater Antilles (13 extinct species, ERT = 8 Myr)[22] and for the terrestrial bird fauna of New Zealand (30 extinct species, ERT = 50 Myr)[21]. It has previously been suggested that the early episode of Holocene extinctions on Madagascar was moderate compared to other islands in terms of its speed (many now-extinct species persisted for notable periods after human arrival), numbers of species and proportion of biota lost[4,49]. The moderate impact of humans on Madagascar compared to other isolated systems (e.g. Caribbean islands[50]) can be explained in part by three different factors that reduced extinction probabilities: (1) larger continental-like area, which allows for higher population sizes[51]; (2) based on the archaeological record of the first millennia after human colonization, expansion was gradual[5] and certain regions have never had important population densities and associated anthropogenic pressures[4] and (3) the vast majority of the island's non-volant mammals are forest-dwelling species[52] and until recent times important areas of forest cover remained intact in most ecosystems[9].

The moderate initial episode of extinctions on Madagascar stands in stark contrast to the current conservation situation. If the many species of Malagasy mammals that are currently threatened go extinct

(5 bats species, 123 non-volant terrestrial species), it would take on average 23.3 (18.5–28.3) Myr (non-volant mammals) and 2.9 (2.3–3.6) Myr (bats) to recover this diversity. In fact, no other insular fauna has such numbers of mammalian species under threat as Madagascar[8]. These ERTs far exceed those reported for the currently threatened birds of New Zealand (13 threatened species, ERT = 6 Myr)[21]. We did not find evidence for temporal rate variation (models M31 and M32, Table S3) and we thus estimated ERTs assuming the average rates across the entire history of the island in the absence of humans. Models with multiple temporal shifts in rates or with other forms of temporal variation are yet to be developed, but we nevertheless favour our use of average natural constant rates (without anthropogenic interference) as this allows for standardization when comparing with the ERT for islands in different biogeographical contexts.

The ERT provides a valuable theoretical measurement of the impact of humans in natural systems considering the unique characteristics of a given island or region. One potential use of the metric is to establish conservation priorities. For instance, if the ERT is estimated for many islands worldwide, we would be able to identify those at risk of losing more evolutionary history[21]. To date, the ERT has been estimated for only two other systems, Caribbean bats[22] and New Zealand birds. In the absence of many other points of comparison, the magnitude and implications of the ERT estimates for Madagascar can be difficult to evaluate. As the ERT values we present are likely underestimates, we can state that the time to recover diversity will not be shorter than the values presented herein, as speciation and natural colonization rates are unlikely to increase, and natural extinction rates unlikely to decrease. Arguably, one of the main conclusions of this study—that the impact of humans on the island will be more severe in the future than in the recent past—would be evident had we just counted threatened and extinct species numbers. Yet, we have demonstrated that species diversity is not a good surrogate for the ERT, which implies the metric provides additional unique information on regional species assemblages. In addition, the ERT provides unique insight into the temporal rewards from conservation efforts, which extend beyond maintaining species richness[53].

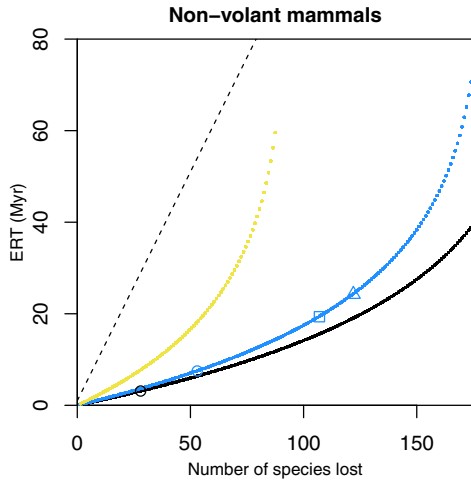
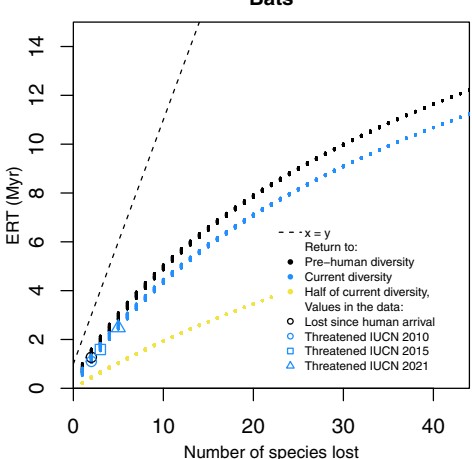

**Fig. 4 | Time it takes to recover species at different stages of the island diversity curve.** Relationship between ERT and number of species to be recovered based on 10,000 simulations with different starting diversities (i.e. assuming different numbers of species have gone extinct) of the best overall model (M26). Curves are shown for different target diversities, to visualize how the time to recover species varies at different stages of the island diversity trajectory.

Our simulations reveal the ERT provides additional insight beyond metrics of human impact based solely on the number of species lost. For example, on Madagascar, at late stages of the diversity curve (e.g. when the island already has a rich mammalian fauna), the ERT per species is larger for bats than for non-volant mammals (Fig. 4). In other words, it takes longer to recover bat diversity than non-volant mammal diversity. This may seem counterintuitive given the much higher rates of colonization for bats, but two factors explain this difference. First, in contrast to non-volant mammal diversity that continues to increase under macroevolutionary scenarios, the island's bat diversity is near a diversity equilibrium plateau (Fig. 4). Second, for non-volant mammals, cladogenesis is considerably higher than extinction and colonization is almost negligible, but for bats the rate of natural extinction is much higher than for non-volant mammals and considerably higher than the rate of cladogenesis. Thus, bat diversity increases via colonization or speciation are balanced out by frequent extinctions. Therefore, because of contrasting macroevolutionary dynamics between the two groups, the loss of a single bat species has consequences that in evolutionary terms outlast those for non-volant mammals.

Madagascar stands out among most other large non-offshore islands because a relatively high proportion of its mammalian biota survived the impacts of human colonization and associated ecological changes[29,54]. However, our analyses of the change in mammalian ERT over the last 10 years on Madagascar reveal a rapid increase in ERT for threatened species, a proportion of which can be attributed to the deterioration of natural habitats and other human pressures, which in turn raise the threat status of species[8]. By assuming species diversity will change at the same rates as in the recent past, our ERTs are likely underestimates, as human impact may have permanently altered aspects of the island's ecology. In addition, the discovery of new species (either already extinct, or extant which are likely to be threatened) could augment the ERT further under some scenarios (Fig. S4).

Increased levels of human activity[10] may eventually lead to the irreversible loss of many of Madagascar's unique threatened species in an extinction episode that would have a much more profound macroevolutionary impact than the one that followed initial human arrival, at least with regards to the non-volant mammal species (Figs. 2, 3, S2, S3, Table S7). We have shown it would take millions of years of evolutionary time for Madagascar to return to its natural pre-human diversity in species numbers, but with adequate conservation policy and action, the loss of the results of many more millions of years of

unique evolutionary history of the island's mammals could be prevented. Based on our results alone we do not recommend prioritizing conservation management of one clade over the others. Given the taxonomic and trait diversity of mammals on Madagascar, the size of the island, high levels of habitat heterogeneity and the fact that all of the native mammal species are endemic, many with very limited geographic distributions, future studies could aim to identify and recommend particular clades and regions that should be the focus of heightened conservation activities to minimize an incipient wave of extinctions. In any case, conservation programs should include socioeconomic improvement for the Malagasy people with particular emphasis on rural livelihoods, reduction of forest loss in the remaining natural habitats and a retooling of the protected area network including better control of artisanal and commercial resource exploitation, including hardwoods and animals for the bushmeat trade[55,56].

## Methods

### Geographical and temporal setting

Paleogeographic and biogeographic evidence suggests Madagascar has been an isolated insular unit since the splitting from Greater India around 88 million years ago (Mya)[1,57]. However, a worldwide mass extinction event is known to have taken place around the K-Pg boundary -66 Mya affecting Madagascar's biodiversity[2], so using this event as an island age for the current biota may also be appropriate. While Madagascar had a rich Mesozoic vertebrate fauna, including many mammals[58,59], molecular phylogenetic data suggest that no mammalian lineage that colonized before the K-T event has survived until the present, although the confidence intervals for the colonization time of the Malagasy Afrosoricida extend until before this age (Fig. 1). Furthermore, the existence of short-lived land-bridges connecting the island to continental Africa at various stages has been proposed[60], although this hypothesis has been contested[61]. We accounted for the uncertainty in island age in our sensitivity analyses, using both 88 and 66 million years (Myr) as island age. We do not consider the possible existence of temporary land-bridges, as we estimate average rates (e.g. of colonization) throughout the entire history of the island (see "DAISIE models" section).

### Malagasy mammalian diversity

We compiled a comprehensive taxonomic/phylogenetic dataset of the entire assemblage of Malagasy non-marine mammals, including information on phylogenetic relationships, timing and causes of

extinction and levels of threat (Supplementary Data S1). We first compiled a checklist of all mammalian species known to have been present on Madagascar in the late Holocene before human arrival and expansion, ~2500 years ago (Supplementary Data S1). The checklist includes all native species that are still extant today and all those that are known or presumed to have gone extinct in the late Holocene. Taxa known only from ancient early- or pre-Holocene fossils that are assumed to have gone extinct long before human presence were excluded, as well as all non-native species. We followed the taxonomy and nomenclature of the Mammal Diversity Database of the American Society of Mammalogists[62,63], as of May 2022. Our checklist does differ from that database because of ongoing taxonomic and nomenclatural revisions based on recent molecular phylogenetic analyses, and recent discoveries or descriptions of new species on the island. All such cases and cases where our taxonomy or nomenclature differs from that used in the phylogeny of[43] are explained in the column "Taxonomy note" in Supplementary Data S1. Molecular-based taxonomy and morphological taxonomy (such as that from the palaeontological record) can be incongruent because cryptic species "detected" in DNA-based analyses may not be identifiable based on fossil data. The number of species in the existing fossil record is therefore likely an underestimate—we address this in the section "Impact of increased knowledge on the ERT".

To compile the checklist, we used a variety of sources from both the neontological and palaeontological literature. The main sources on Malagasy extant and extinct mammalian species are[4,8,25,29,52,57,58], but other published studies were used, particularly for bats, lemurs and cryptic and recently discovered species (Supplementary Data S1). We classified species as endemic or non-endemic to Madagascar, as information on endemicity status is one of the types of data that DAISIE uses to estimate rates of speciation. For non-endemic species, only represented in the dataset by certain species of bats, we noted their range outside of Madagascar in the column "Additional range note". For the ERT analyses, we also compiled the IUCN Red List status for each species in 2010, 2015 and 2021. We used the digital archive "wayback machine" to obtain the 2010 and 2015 IUCN Red List data, as older versions of listings are not kept online by the IUCN[8,41,42]. We compiled the checklist using Excel v16.63.

### Extinct species

In the DAISIE analyses we treat species that went extinct due to non-anthropogenic causes (before or after human arrival) as if they were species not known to science. This is because the natural extinction rate that is estimated based on the colonization and branching times extracted from phylogenies (without these extinct species), already accounts for such missing species. These include species that went extinct before the late Holocene and may or may not be known from the fossil record, but also species that have gone naturally extinct after human arrival. In contrast, we consider that anthropogenic extinctions do not contribute to the natural extinction rate. Therefore, we treat species for which an anthropogenic cause of extinction is likely as if they had survived into the present and we include them in the phylogenies, following the approach of Valente et al.[22]. The rates of speciation, colonization and natural extinction that are estimated from such phylogenies are the natural average rates assuming that humans had no impact on the island—these would be the natural average background rates in the periods pre-dating human arrival.

Our checklist includes all species that are hypothesized to have gone extinct in the late Holocene, hereafter termed "recently extinct species". For these species, we compiled information on whether they have gone extinct before or after human arrival (columns "Last date $^{14}$C age BP" and "Extinction before/after humans" in Supplementary Data S1). In assessing whether species went extinct before or after human colonization and expansion, we employ the circa 2500 years BP date as time zero[3–5,64,65]. After permanent settlement, anthropogenic

pressures on Madagascar's biodiversity have intensified, with a visible increase in the past few decades[4]. Species which are considered in the literature to have gone extinct before human arrival were excluded. For some taxa, it is unclear whether they went extinct before or after human colonization, because the fossil record is insufficiently known. These were included in the list and the effect of their inclusion/exclusion was evaluated in sensitivity analyses (see section below). In addition, for all extinct species in our list, we compiled information on the hypothesized causes of extinction cited in the literature, classifying each extinction as anthropogenic, natural or uncertain. References for timing and causes of extinction are provided in Supplementary Data S1.

To account for these uncertainties, we re-ran analyses for two datasets, assuming high and low human impact (see sensitivity analyses). For the high human impact scenario, we assumed all recent extinctions (less than 2500 years ago) to have an anthropogenic cause and therefore included them in the phylogenies and in the counts of pre-human species diversity. In this scenario we also include recently extinct species for which no $^{14}$C dates are available, but which have been hypothesized to have gone extinct in the last 2500 years, as well as species with a putative natural cause of extinction, because even natural recent extinctions may have had an indirect human influence. For the high human impact scenario, we thus assume all recent species loss is linked to potential human influence, and we include all those species in the phylogenies. In the low human impact scenario we assume a natural cause for all recent extinctions that have previously been hypothesized to have had a natural cause, for all recent extinctions whose cause is unknown, and for the cases for which it is unclear whether extinction pre-dates or post-dates human arrival (e.g. no radiocarbon dating available). For the low human impact scenario, we excluded all such species from the phylogenies and from the counts of pre-human diversity (it is assumed they are unknown). See column "Low human impact scenario" in Supplementary Data S1. For two of the species for which no date currently exists indicating whether they went extinction before or after humans (the lemurs *Mesopropithecus dolichobrachion* and *Palaeopropithecus kelyus*), an anthropogenic cause of extinction has been hypothesized despite the lack of a precise last occurrence date, and we thus considered them to be anthropogenic extinctions in both high and low human impact scenarios.

### Phylogenetic data

The source of all our phylogenetic data—including divergence times of Malagasy lineages—is the phylogeny by Upham et al.[43], the most comprehensive and complete mammalian phylogeny published to date, including 5911 species of mammals. From this tree, we extracted phylogenetic information with reference to Madagascar on the number of colonization events, the estimated dates of colonization (divergence times from the most closely related non-Malagasy relatives), number of species per monophyletic colonist lineage and the timing of within-island speciation events. We created a Madagascar-specific dataset consisting of a series of multiple subtrees drawn from the same Mammalia-wide dating framework, representing all colonization events for most known late Holocene native mammals on the island, including bats and recently extinct species. We visually inspected the trees using Figtree v1.4.4.

Upham et al.[43] used two approaches to calibrate their phylogeny: node dating and tip dating. Following the recommendations in that publication, we used the trees based on the node-dating approach, in which node-age priors were placed on the tree based on 17 mammalian fossils and one root constraint. Regarding molecular sampling, they produced two types of trees: DNA-only, with 4098 species sampled in the phylogeny based on molecular data; and completed trees, where they placed an additional 1813 species that were unsampled for DNA in the tree using taxonomic constraints (across multiple posterior trees). The DNA-only trees have the advantage that the topology is based on

molecular data, and is likely more reliable, but the disadvantage that DNA sequences were not available for many Malagasy species and so these species needed to be added to the phylogeny for the DAISIE analyses. The completed trees have the advantage that they are near-complete, but the disadvantage that some Malagasy species—particularly several bats that are unsampled for DNA—were placed randomly within a given clade constraint, which may lead some trees in the posterior to have some incorrectly inferred colonizations. We ran analyses on data extracted from both types of trees (see sensitivity analyses).

An alternative to using this phylogenetic dataset would be to extract data from separate individual trees from publications with phylogenies focusing on specific clades. There are many such studies, and indeed some of them include taxa that are not present in the Upham et al.[43] tree—e.g. new recently described cryptic species that were only identified after molecular analyses, including, for example, the nesomyine rodent *Eliurus tsingimbato*[66] and the mouse lemur species *Microcebus jonahi*[67]; or extinct species for which no molecular data exists, but which were included in phylogenetic dating analyses based on morphological data, such as members of the lemur genus *Mesopropithecus*[46]. However, we favoured using phylogenetic data from a single study to ensure divergence times are comparable (i.e. same models, assumptions and data), even though this is done at the expense of reduced species sampling. Although we use a single tree (or posterior distribution of single trees) for our dataset, DAISIE treats each Malagasy colonizing lineage as its own separate tree, so we deal with a "forest" of phylogenetic trees, each representing a single Malagasy lineage resulting from one colonization event. For example, the lineages that have radiated on the island have a tree that includes the stem age of the lineage (splitting from the closest sampled mainland relative) and all branching events within the radiation. Lineages with a single species on Madagascar (endemic or non-endemic) are essentially a tree with a single tip and with an age equal to the splitting of that species from its closest (sampled) continental relative.

### Alternative colonization scenarios

The number of colonization events of Malagasy mammals inferred from the phylogenetic data can vary depending on the placement of some missing taxa in the tree or because some clades have poor branch support and could be the result of one or more colonizations in different trees from the posterior. We considered two alternative colonization scenarios (CS), one where we favour fewer colonizations (CS1) and one where we favour more (CS2). The differences between the two scenarios are summarized in Table S1 (all colonizations shown in Supplementary Data S2, S3). We considered lemurs to be the result of a single colonization event in both scenarios. A recent study[68] has suggested that *Daubentonia* is a separate colonization of Madagascar, but we assigned both the extant aye-aye and the extinct giant aye-aye to the single Lemuroidea (lemurs) clade because that is the only scenario supported by the mammal tree.

### Adding missing species

A total of 34 out of 249 species in our Madagascar mammal checklist are not present in the mammal phylogenetic tree[43]. Most of these (23 species) are extinct species (Data S1). The other 11 species are recently described species (two bats, eight lemurs and one nesomyine rodent), these are indicated in column "Taxonomy note" in Supplementary Data S1. An additional 61 species are included in the completed trees, but not in the DNA-only trees, as no molecular data were available for these. We added the 34 species missing from the mammal tree to both DNA-only and completed trees, and the 61 species missing molecular data to the DNA-only trees. Instead of adding species directly to the posterior distribution of trees and then extracting information from the phylogenies, we assign those species to specific clades using the "missing species" option in DAISIE. This tool allows

them to be placed anywhere within the Malagasy clade they are believed to belong to, without specifying a specific topological position within the clade—DAISIE does not use topological information for its estimates. For example, a species of lemur that was missing from the tree was added to the species count of the lemur clade. The information on the clade to which each missing species was added to (under either CS1 or CS2) is provided in Supplementary Data S1.

Most recently extinct species are not included in the mammal tree because the original study was primarily focused on the extant mammalian taxa[43]. Three extinct species of lemur, *Archaeolemur majori*, *Megaladapis edwardsi* and *Palaeopropithecus ingens* are included in their tree based on molecular data obtained from subfossil material. One extinct species of lemur (*A. edwardsi*), one extinct species of carnivoran (*Cryptoprocta spelea*) and two extinct hippopotamus species (*Hippopotamus madagascariensis* and *H. lemerlei*) are included in their completed trees, i.e. not based on molecular data. We added the remaining extinct species to the phylogenies using the approach explained above (Supplementary Data S1). These were: 1 tenrec (assigned to the Malagasy Afrosoricida (CS1) or Tenrecidae (CS2)); the 2 bibymalagasy (assigned to Malagasy Afrosoricida (CS1) or to Bibymalagasia (CS2)); 1 hippopotamus (assigned to the single hippopotamus clade (CS1) or to one of the two hippopotamus clades (CS2)); 1 euplerid carnivore (*Cryptoprocta* sp. nov., assigned to Eupleridae); 2 bats (1 *Paratriaenops*, assigned to the *Paratriaenops* clade; 1 *Macronycteris* assigned to *Macronycteris* (CS1) or as its own colonization (CS2)); 3 nesomyine rodents (assigned to Nesomyinae); and 13 lemurs (assigned to the Lemuroidea clade).

In a few cases, all descendants from a colonization of Madagascar were missing from the mammal tree. These were added as a separate colonization, using the DAISIE_max_age option, which assumes that they could have colonized at any time since the given age and the present. These were: the two species of Bibymalagasia (CS2, using the stem age of Afrosoricida in the mammal tree as the maximum age of colonization); *Chaerephon leucogaster* (CS1 and CS2, using crown age of Molossidae family as maximum colonization time); *Macronycteris cryptovalorona* (CS1 and CS2, using crown age of Hipposideridae family as maximum colonization time); *Macronycteris besaoka* (CS2, using crown age of Hipposideridae family as maximum colonization time); *Hippopotamus laloumena* (CS2, using stem age of genus *Hippopotamus* as maximum colonization time); and *Pipistrellus raceyi* (absent from the DNA-only tree, we used the crown age of Vespertilionidae as the maximum colonization time). For *Miniopterus*, the phylogenetic resolution for this radiation is poor (including both Malagasy and non-Malagasy taxa), and we therefore used the crown age of the genus as a maximum colonization time of Madagascar (we chose the crown and not stem because *Miniopterus* of Madagascar do not diverge early in the genus).

### Colonization and branching times

For endemic Malagasy clades (radiations (e.g. lemurs) or clades with a single endemic species, e.g. *Pteropus rufus*), we assumed the time of colonization of Madagascar coincides with the divergence time from its closest non-Malagasy lineage, i.e. the stem age of the clade. These ages are likely overestimates (e.g. if the tree is incompletely sampled, or if the closest continental ancestor has gone extinct, see ref. 18), but are a good approximation, and we repeated analyses over the posterior distribution of trees to account for age uncertainties. Non-endemic species are represented by a single tip in the mammal tree, and we therefore used the age of that tip as a maximum age of colonization, as the actual colonization time of the Madagascar population is most certainly younger than that age. DAISIE integrates through all possible ages between that maximum age and the present. The only exception are three non-endemic species belonging to the Chiroptera genus *Miniopterus*, which likely resulted from cladogenesis within Madagascar and became non-endemic by colonizing the Comoros.

These are treated as part of the *Miniopterus* clade or clades (Table S1), and thus contribute to the estimates of cladogenesis on Madagascar (rather than being assigned their own colonist lineage). The branching times within Madagascar radiations were taken directly from the trees. When species within a radiation were missing from the phylogeny, they were included using the DAISIE missing species option (see section above), thus contributing to the estimates of cladogenesis rates for the given clade.

We wrote an R script to extract colonization and branching times from the maximum clade credibility (MCC) and posterior trees from the Upham et al.[43] mammal phylogeny, add missing species and assign maximum colonization times (if relevant), assuming a variety of scenarios (see "Main dataset and sensitivity analysis" section below). Once the data were extracted from the trees, the script creates DAISIE objects, i.e. datasets in DAISIE format that can be read by DAISIE functions. This script uses functions from phytools v1.2-0 R[69], ape v5.6-2[70] and DAISIE[47] v4.0.5 R packages. The R script describes all the steps taken to prepare the phylogenetic data for the DAISIE analyses. The script, the precise source trees that we used from the Upham et al. mammal phylogeny[43], as well as all DAISIE objects for the main analyses and the sensitivity analyses are provided in an online repository (https://github.com/luislvalente/madagascar). Analyses were run in R v4.2.1 and RStudio v2022.02.3.

## DAISIE

We used the DAISIE R package[47] to estimate rates of speciation, colonization and extinction (CES rates) of Madagascar mammals using maximum likelihood (ML) under a range of different models and to identify the preferred model given the phylogenetic data. The DAISIE likelihood inference approach is based on theory and methods developed for phylogenetic birth-death models[71,72]. It has been demonstrated that the shape of phylogenies of extant species contains information about natural extinction rates[72]. While these approaches have many known limitations[73], we have shown in different studies that the DAISIE model is able to accurately estimate extinction rate from simulated datasets for which the extinction rate is known[18,74]. In addition, unlike most phylogenetic birth-death models, which are single-clade approaches and use only information from branching times, DAISIE has the advantage that it uses information from multiple independent clades and from both colonization and branching times, increasing its statistical power to estimate parameters.

We fitted a set of 30 DAISIE models to the phylogenetic data, explained in Table S2. Models M1–M4 assume homogeneous CES rates for all Malagasy mammals, while for models M5–M30 we allow one or more of the CES parameters to vary between non-volant mammals and bats. The set of models include both diversity-dependent and diversity-independent models. Models can differ in the number of parameters: for example, M1 has five parameters (colonization, cladogenesis, anagenesis, $K$ and extinction); M3 has four parameters (same as M1, except that anagenesis is fixed to zero); M5 has six parameters (the same five parameters as M1, plus a parameter for colonization rate which differs for bats); and M6 has five parameters (same as M5 but $K$ is fixed to infinite, i.e. there is no diversity dependence).

## CES rate heterogeneity

DAISIE estimates average CES rates for the island and assumes that these rates are constant through time (except for models that include diversity dependence, in which rates decline with increasing diversity). However, from the geology of the island and the fossil record, we can infer that rates have most likely not been constant. For example, periods of large-scale natural extinction may have taken place throughout the history of the island[3,75,76]. While there may have been important temporal rate changes, when estimating the future island evolutionary return time (the main purpose of our analyses), we seek to estimate the overall average natural background rates, which incorporate periods of both low and high rates (which will certainly also occur in the future). We fitted two models (M31–bats and non-volant mammals share same rates; and M32–bats and non-volant mammals have different rates) in which colonization rates can shift to a lower or higher rate[46] at a certain point in time, but these models were not preferred (Table S3). Therefore, when we estimate the ERTs, we use the average rates for the island as a whole over its entire geological history in the absence of humans. Importantly, although the preferred models assume constant rates, the DAISIE model has been shown to perform very well for ancient continental islands (separated from the mainland very deep in geological time, such as Madagascar), in terms of accurately predicting the number of species, and the number of species and colonizations through time[77]. In addition, although rates may have been lower or higher at some periods, the average rates are nevertheless informative of the unique geographical setting of the island and the ecological characteristics of the target community—this is particularly valuable, for example, when comparing Malagasy mammals with ERTs from other systems, such as in Caribbean bats[22] and New Zealand birds[21], both in which rates have also most likely varied through time.

There is evidence for rate variation among mammalian lineages[78]. We therefore chose to test for differential rates for two groups: non-volant mammals and bats. In the context of islands, it is likely that bats will have different rates of colonization due to their higher dispersal abilities, and they may also vary in other parameters. We used the two-type DAISIE model approach first applied to the birds of the Galápagos (Darwin's finches vs other birds[45]. While there may also be differences in rates between specific non-volant and bat clades, we favour obtaining average ERTs across the whole fauna, rather than specific ERTs for each lineage. First, assigning unique rates to each lineage would lead to over parameterization, and estimating lineage-specific rates would not be reliable for some individual Malagasy clades that are the product of a single colonization and have few species (e.g. many bat lineages, hippopotamuses, euplerid carnivorans). Thus, we restricted the test of idiosyncrasies to the comparison between bats and non-volant species. Second, an advantage of our approach is that the rates we obtained are based entirely on the phylogenies of Malagasy species and therefore our rates are already very specific to the Malagasy context—whereas comparable methods use average rates worldwide and then extrapolate to the focal lineages[24]. Third, we are interested in whether total diversity will recover, not whether specific types of species will recover. A trait-dependent diversification model for insular communities that would allow us to obtain ERTs based on, for example, certain morphological traits that may promote diversification, does not currently exist.

## Main dataset and sensitivity analyses

We consider the main dataset (D1) to comprise: colonization scenario 1 (CS1) with high human impact, using the DNA-only mammal tree (MCC and posterior), island age of 88 Myr. The reason for this is that we consider the CS1 (fewer colonizations) and high human impact scenarios to be the most realistic given the level of isolation of the island and because the evidence for anthropogenic mammalian extinction on Madagascar is compelling and growing[6]. We also consider the DNA-only tree more appropriate, as species were sampled based on molecular data, and all missing species were included in clades using the DAISIE missing species option, i.e. placing them in a clade but without forcing a given topology within that clade.

There are currently ~6500 species of mammals[62], but this number was certainly different in the past and only a subset of these constitute the potential mainland pool for Madagascar, which would include African species and to a much lesser extent from the Indian subcontinent or other portions of Asia. For the main dataset we considered the number of species on the mainland pool (M) to be 1000

(approximately the current number of African terrestrial mammal species), but we re-ran analyses with 2000 and 5000 species. For the models where bats differ from non-volant mammals (M5–M30), the proportion of bat species in the mainland pool was set to 0.22, equivalent to the proportion of all mammal species that are bats today.

To account for uncertainty in island age, mainland pool size, colonization scenarios, human impact, topology, dating (colonization and branching times), and tree sampling completeness, we ran a series of sensitivity analyses. We re-ran analyses for the MCC tree of the main dataset (that is: DNA-only tree, high impact, CS1), assuming an island age of 66 Myr, and varying pool sizes (for both island ages). Then, fixing the mainland pool to 1000 species and the island age to 88 Myr, we ran DAISIE analyses assuming colonization scenarios CS1 and CS2, high and low human impact. We also repeated analyses using the completed and DNA-only trees, using the corresponding MCC trees for each scenario. In total we ran 13 different scenarios (D1–D13) for the sensitivity analysis (Supplementary Data S4).

We used the following approach for ML optimizations on the main dataset and the sensitivity analyses, using the DAISIE_ML function implemented in the DAISE R package. For the analyses on a single MCC tree (all 13 scenarios, including the main dataset), we fitted each of the 30 DAISIE models to each dataset 10 times, using different random sets of starting values for the likelihood optimization ($30 \times 10 = 300$ ML optimizations per scenario, total 3900 ML optimizations). For each scenario, we selected the preferred model by comparing Bayesian information criterion (BIC) and Akaike information criterion (AIC) scores between models. For the main dataset, to examine if the same model is preferred across the posterior distribution of trees, we also ran analyses on the posterior, fitting each model 4 times to each of 100 datasets from the posterior ($30 \times 4 \times 100 = 12,000$ optimizations). To obtain confidence intervals for the preferred model of the main dataset, we ran analyses on 1000 trees from the posterior, with 2 random sets of starting values ($2 \times 1000 = 2000$ ML optimizations, Table S4). We ran ERT analyses using the parameters of the preferred model for all 13 scenarios. For the main dataset D1 we also fitted two models with a temporal shift in colonization rate (M31 and M32, Table S3). All analyses were run on the Peregrine cluster of the University of Groningen.

For the main dataset, we ran simulations of the best overall rate model using the DAISIE_sim function. Under the parameters of the model, we simulated 5000 islands for 88 million years. We then assessed the goodness of fit of the model to the data by comparing diversity metrics in the simulated datasets to those in the empirical data.

In the sensitivity analyses, varying mainland pool size, island age, human impact, colonization scenario or phylogenetic dataset (DNA-only vs completed), had a limited impact in the preferred models or parameters values (Supplementary Data S4). Varying island age, human impact or colonization scenario generally led to only minor changes in parameter values. Varying mainland pool sizes affected the colonization rate, which decreases with mainland pool size because colonization rate is measured per mainland species. When using BIC as the criterion for model selection, M26 was the preferred model in 10 out of 13 scenarios, with M22 being the preferred model under one scenario (D8, DNA-only data, island age 88, M = 1000, CS2, high human impact), and M11 preferred under two scenarios (D12 and D13, completed trees, island age 88, M = 1000, C2, for both high and low human impact) (Supplementary Data S4). Using AIC, alternative models to M26 were preferred for two additional scenarios - M10 was preferred for D10 (completed trees, island age 88, M = 1000, CS1, high human impact) and M11 was preferred for D11 (completed trees, island age 88, M = 1000, CS1, low human impact). We consider the M26 model to be the preferred model overall, because we favour the DNA-only trees (for which M26 was consistently selected as the best model under both AIC and BIC) and the BIC criterion for model selection (shown to perform

better than AIC when selecting between DAISIE models[18]). However, like M26, all three alternative models preferred in some of the sensitivity analyses (M10, M11 and M22) are two-rate models under which bats have a higher rate of colonization than non-volant mammals and differ from non-volant mammals in one or more parameters.

## Preferred model

The preferred model is the M26 model. Under this model, the background rate of cladogenesis for Malagasy mammals is 0.33 (0.27–0.36) events per lineage Myr$^{-1}$ and the rate of anagenesis is 1.47 (1.18–2.12) events per lineage Myr$^{-1}$ (Table S3, S4). The model is diversity independent (for both bats and non-volant species), meaning that there is no carrying capacity per clade (K per clade is infinite). The rate of natural extinction for non-volant mammals is 0.29 (0.22–0.31) events per lineage Myr$^{-1}$, and for bats it is 0.46 (0.40–0.50) species per lineage Myr$^{-1}$. The rate of colonization for non-volant mammals is 0.00036 (0.00027–0.00038) events per mainland species Myr$^{-1}$, equivalent to 0.28 colonizations per Myr (0.21–0.30). The rate of colonization for bats is much higher, at 0.034 (0.030–0.038) events per mainland lineage Myr$^{-1}$, equivalent to 7.5 successful bat colonization events per Myr (6.6–8.4).

## Evolutionary return times

The island evolutionary return time (ERT) metric estimates the time it would take for an insular community to reach a given species diversity level assuming a given model of macroevolution with certain rates of colonization, speciation and natural extinction[22]. To estimate ERT, we first counted the number of species that were present on Madagascar in the late Holocene (pre-human diversity) (Table S5). We also counted the number of mammal species estimated to remain extant on the island if currently threatened species go extinct. Threatened species we classified as those that fall under the IUCN categories Vulnerable (VU), Endangered (EN) or Critically Endangered (CR). We estimated the ERT of Malagasy mammals for the following scenarios: (1) the return from current diversity to pre-human diversity, and (2) the return from the diversity that will remain if threatened species (VU + EN + CR) go extinct back to current diversity.

There were some differences in ERTs between datasets (D1–D13). Most differences were subtle and some appear counter-intuitive (Table S6). For example, a lower number of colonization events in the data (CS1 vs CS2) or a larger mainland pool both lead to a lower colonization rate, which evidently should increase the ERT. However, changes in the mainland pool and the number of colonization events can also lead to very small changes in the diversification rate (cladogenesis minus extinction), which has a much higher impact on the ERT than colonization rate. The largest differences in ERT were between the high and low human impact scenarios, because not only do estimated rates differ, but also the start and target diversities.

## Change in ERT between 2010 and 2021

To compare how ERT for threatened species has been changing through time as human impact and working knowledge increases, we repeated analysis (2) using the IUCN threat statuses from 2010, 2015 and 2021. As the threat status of some species increases, they are uplisted by the IUCN, e.g. from Near Threatened (NT) to Vulnerable (VU) (e.g. the Madagascar rousette (*Rousettus madagascariensis*)), i.e. becoming threatened under IUCN classification of threat (VU, EN, CR). Although there have been many changes in status between the categories that we considered as threatened (e.g. from VU to EN, or from EN to CR), these are not considered in our analyses, as we only consider changes from non-threatened to threatened. Therefore, our analyses can be considered conservative; hence, the increase in ERT from 2010 to 2021 is probably higher than what we estimate. It is important to keep in mind that

for different reasons, which include more intensively studied Malagasy mammalian groups and biopolitics, the manner certain data are weighed during assessments and resulting statutes are not necessarily the same across mammal groups.

A total of 72 species were uplisted by the IUCN from a non-threat to a threat category between 2010 and 2021 (Supplementary Data S1). Of these, 57 moved from Not Evaluated (NE) or Data Deficient (DD) categories to one of the threat categories (VU, EN, CR), so may simply represent an increase in knowledge rather than a real increase in threat status. In addition, IUCN categories are not comparable across assessments for species that have undergone taxonomic revisions that may have altered their threat status−e.g. splitting of one species into two allopatric species will lead to a range size reduction. We therefore also repeated analyses only for those species that changed from an evaluated non-threat category (LC, NT) to a threat category (VU, EN, CR) and which have not undergone taxonomic changes between 2010 and 2021 or for which a taxonomic change was not the cause the up-listing. For all species that changed from a non-threat to a threat category between 2010 and 2021 (15 species, Supplementary Data S5) we consulted the literature to find out whether changes in taxonomy took place for that taxon, and whether those changes influenced the up-listing. We identified 10 species (Supplementary Data S5) for which there was no taxonomic change between 2010 and 2021 or for which a taxonomic change did not lead to an up-listing between those years. We then calculated ERTs for a scenario where we assume that only those 10 species were uplisted between 2010 and 2021 (Table S8, scenario C).

### Species diversity lost versus ERT

The number of species lost through anthropogenic extinctions and the ERT are two alternative ways of looking at the impact of humans on island biota. To assess whether the number of species lost is a good proxy for ERT, we ran simulations to measure how ERT varies with the number of extinct species to be recovered. For example, we compared how the time to return to pre-human diversity varies with starting diversity, e.g. assuming an increasing number of species have gone extinct. We ran simulations using the parameters of the best overall model for the main dataset. We first created 10,000 random start diversities, sampling between 0 species and a target species diversity, assuming variable numbers of species have gone extinct. For each of the starting diversities, we randomly specified a proportion of endemic and non-endemic species. Simulations were run in R.

The shape of the island species diversity curve (how the total number of species on an island varies through time) under different DAISIE models can vary at different stages. For example, in an equilibrium model, diversity increases rapidly at early stages and low diversities, but at later stages it plateaus and increases slowly. In non-equilibrium models, diversity increases can for example be low at early stages and faster later. This will have implications for how ERT relates to the number of species that need to be recovered. We therefore separated the results into (a) returning to pre-human diversity (capturing a later stage of the diversity curve), (b) return to contemporary diversity (capturing an intermediate stage) and (c) return to half of the contemporary diversity (capturing early stage of the diversity curve). We did this separately for non-volant mammals and bats.

### Impact of increased knowledge on the ERT

New species discoveries and increasingly complete IUCN Red List assessments are likely to affect ERT estimates in the future. The discovery of new extant species may lead to an increase in ERT because undiscovered species are more likely to already be threatened (e.g. due

to small range and population sizes). Taxonomic revisions may lead to species splits, resulting in additional threatened species. The known fossil record may also include cryptic species that cannot be identified using molecular methods if DNA is not available. The discovery of more taxa that have gone extinct since humans arrived will also likely increase the ERT (return to pre-human diversity). However, if new species are discovered, the rates of colonization and speciation estimated in DAISIE will also increase, and therefore ERTs may not rise dramatically.

To assess how future species discoveries may affect our results, we performed analyses where we assume 30 new mammal species (15 bats species and 15 non-volant mammal species) will be discovered in the next 10 years on Madagascar. This is likely an overestimate. We simulated datasets by adding these bat and non-volant species at random locations to the main phylogenetic dataset D1, and repeated this procedure 1000 times. We fitted the preferred DAISIE model to these 1000 datasets and estimated the ERT for each of them. We then assumed that the newly discovered species were (a) all threatened, (b) half of them threatened and half already extinct or (c) all already extinct (since human arrival). The results of these analyses are summarized in Fig. S4. We found that the ERTs for non-volant mammals do not change substantially, increasing slightly or even declining under some scenarios. This is because although the number of species to recover increases, the estimated DAISIE rates also increase. On the other hand, under some scenarios, an increase in the number of extinct or extant bat species leads to large increases in ERT, as it takes longer on average to recover bat species according to the preferred DAISIE model.

IUCN Red List assessments should become more comprehensive in the future; currently 8% of recognized Malagasy mammal species are Not Evaluated or assessed as Data Deficient, corresponding to 18 species (10 bats and 8 non-volant species). Thus, we also estimated how the completion of IUCN assessments may affect our results. If all species yet to be assessed by the IUCN were evaluated as threatened in the future, the ERT to return to contemporary diversity would rise to 26.2 (20.8−32) Myr for non-volant mammals (~13% increase) and 6.6 (5.5−7.8) Myr for bats (more than double) (Table S8, scenario A). The increase is proportionally higher for bats because it takes longer to recover bat species and because there are more unevaluated bat species.

### Reporting summary

Further information on research design is available in the Nature Portfolio Reporting Summary linked to this article.

### Data availability

All the data that support the findings of this study are provided with this paper or have been deposited online. The Malagasy mammal checklist is available in the Supplementary Information. The phylogenetic trees and DAISIE objects are available in a Zenodo repository https://doi.org/10.5281/zenodo.7311466. The posterior distributions of phylogenetic trees are available in Mendeley Data https://doi.org/10.17632/wfs8k2pphd.1. We harmonized the taxonomy based on the Mammal Diversity Database available in ref. 63.

### Code availability

The R scripts used to extract the DAISIE data from the phylogenetic trees, perform the main ERT analyses and reproduce the main figures are available on Github (https://github.com/luislvalente/madagascar) and archived on an open data repository hosted by Zenodo at https://doi.org/10.5281/zenodo.7311466. The R code to run the DAISIE analyses can be found in DAISIE R package v4.0.5 https://zenodo.org/record/5708159#.Y200qy8w19g or and on Github https://github.com/rsetienne/DAISIE.

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

## Acknowledgements

We thank Tom Nijssen, Olle Odijk, Tim Reimes and Dolf Rutten for help compiling the data. Olle Odijk for code used in the ERT figures. L.V. was supported by a Vidi grant from the Dutch Research Council (NWO 016.Vidi.189.006); L.M.D. in part by NSF-DGE 1633299; N.S.U. by the Arizona State University President's Special Initiative Fund and NIH 1R21AI164268-01; L.M.D. and N.S.U.'s portion of the work was supported by the National Socio-Environmental Synthesis Center (SESYNC) under funding received from the National Science Foundation DBI-1639145.

## Author contributions

L.V., S.M.G. and L.M.D. initiated the research. N.M.M. and L.V. designed the study, compiled the data, performed the analyses and wrote the first draft. S.M.G. and V.S. provided expertise on Malagasy mammals and curated the dataset. S.M.G., G.I.S., V.S. and A.E.G. helped compile the dataset. A.E.G., L.M.D. and N.S.U. provided expertise on various aspects of the data, writing and analyses. All authors commented on various versions of the draft.

## Competing interests

The authors declare no competing interests.
