## [Peer Review File · Nature Communications]

The macroevolutionary impact of recent and imminent mammal extinctions on MadagascarREVIEWER COMMENTS

Reviewer #1 (Remarks to the Author):

In their manuscript the authors use an island biogeography model to evaluate the impact of previous and future extinctions of non-volant mammals and bats in Madagascar. I think that the article is well-written, well-structured and, although I am not an expert in macroevolutionary data and models, includes a thorough compilation of phylogenetic data for Malagasy mammals. In the body of the paper, the methods are clear and the results and conclusions follow logically. The paper contributes to understanding the effects of losing species (diversity) which is important in terms of management and policy considerations.

However, there are three aspects of the paper which could potentially be improved and/or on which I believe the authors could provide more detail. First, I understand the reasoning why the authors have used constant CES rates through time but I question the usability of constant rates based on historical data for future predictions. Would it also be possible to use non-constant rates over time? For example, could the authors use rates of more recent times to predict the ERT when all threatened species would go extinct? Recent colonization rates might for example be more suitable for future predictions than rates based on the entire time span, as humans may have (had) a positive influence here. This would, in my opinion, reflect reality better.

Second, related to the first point, I was not sure after the reading the MS in what way colonization exactly contributes to species diversity in DAISIE. Is it correct that colonization could directly lead to higher diversity values when the particular species was not yet occurring (or has gone extinct) on Madagascar? I am then wondering whether the approach should not focus on endemic species only and thus indirect contributions of colonization. In other words, ERT (to me) seems to be more valuable to investigate the global loss of species instead of looking at local diversity. So should you then take colonization even into account apart from the routes it may provide for cladogenesis and anagenesis?

Finally, after reading the MS, I would have a few remaining questions if I was a conservationist regarding the applicability of the results. I think it is very important to know what would be the consequences of losing insular species and (as the authors describe) the ERT can be used to compare different islands to see where effects are most detrimental. But now that we know that losing species on Madagascar will take a long time to recover, what should conservationists do next? Which species and areas on Madagascar should be protected/conserved? I do not expect the authors to answer these questions in the MS but I do feel that these important aspects should be discussed. What in my opinion would help in this regard, is to investigate the (potential) relationship between clade/species characteristics and ERT. In lines 974 - 976 the authors state that a trait-dependent diversification model for insular communities not yet exists, but could the authors not run simulations with different species diversity (and thus different missing species) to find out what species characteristics/lineages prove to be more difficult to replace (similar to the approach described in lines 1120 - 1123)? This would greatly increase the significance of the paper by helping to understand which species are especially difficult to 'replace', and, in turn, which should receive conservation focus (also potentially outside of Madagascar).

Minor points:

-The authors have shown in their MS that the species lost is not a good surrogate for ERT and that the link between the two is non-linear. The assumption made by the authors for future predictions now is that all threatened species will go extinct, no matter the difference in IUCN categories, whereas it is likely that CR species will go extinct earlier compared to VU species. How would this affect the results?

-Lines 64 - 66: to me the interrogative sentences read a bit strangely so I would suggest to rewrite these.

-Lines 98 - 99: I would suggest to remove 'in a conservative manner' here. Based on the previous sentence, I would argue that the authors are not that conservative here as they state that humans "largely" disappeared for 8000 years.

-Lines 118 - 120: suggest to rewrite as an increase in threatened species is not directly related to

ERT or to the number of species lost (only with the assumption made by the authors that all threatened species will go extinct in the near future which has not been stated in the MS yet).

-Lines 757 - 758: I would appreciate if the authors can include the timing of extinction in Supplementary Data S1. This makes it more easy as a reader to follow the assumption made in the MS.

-Line 922: change colonisation to colonization.

-Table S3: please also report the AIC and BIC values here which were used to select the best model.

-Table S6: based on the description of the data sets and the parameters therein, I would assume that: 1) the larger the mainland pool the lower the ER, and 2) CS2 would lead to a lower ERT compared to CS1. This is not (always) the case. Can you explain why? Similarly, why would the ERT for non-volant mammals be higher in D4 compared to D1? The opposite is true for D5 vs D2 and D6 vs D3.

-Lines 1034 - 1035: Related to my previous point, you state here that the colonization rate decreases with mainland pool size. Why is this? Simply because the unit for colonization rate that you used is per mainland species? Then it would be good to specify. Or is it because of homogenization?

-Line 1157: change 'extinctor' to 'extinct or'.

-Figure 1: I would suggest to shade the filled symbols on top of the dashed lines to highlight that these are estimates/random.

-Line 240: bats follow an equilibrium model towards 57 species. How does that relate to a carrying capacity of an infinite number of species in the model? It is surprising to me that this model (with setting K to infinity) then performs better than when the carrying capacity was an emerging property for bats. Can you explain?

-Figure 2: from the figure I conclude that the diversity goals (e.g. pre-human diversity) are set separately for bats and non-volant mammals, right? In other words, when the pre-human diversity for bats of 44 is reached, the additional/new bat species (from the found limit of 57 species) do not contribute to reaching the goal, right? I would do specifically state this in the main text. NB: are figures 2C and 2D needed?

-Line 267: suggest to change to 'only 91 species of mammals out of a total of 219 mammals (current diversity) would remain on the island'.

-Figure 2 and 3: I would suggest to use only one of these figures in the main text as the information they hold overlaps. Note that the authors always also report to both of these figures when discussing the results.

Reviewer #2 (Remarks to the Author):

This manuscript offers a novel perspective on the extinction crisis in Madagascar by using quantitative approaches that integrate speciation times, inferred dispersal events, geological context, and human impacts. Notably, it also focuses on bats which have been under appreciated in conservation analysis in Madagascar and elsewhere. Though I am not familiar with the primary analytical method (DAISIE), the assumptions seem sound.

I believe that this will be an important contribution to the extinction/conservation literature in that it makes the evolutionary costs of human-mediated extinction abundantly clear.

I have a few minor comments for improvement:

- the sentence in the introduction (lines 51 - 55) relating to human impacts is confusing, I think due to the word "protracted." If I understand the meaning of the sentence correctly, it seems like a word like "delayed" would be better suited to the point that biodiversity persists given that human impacts have only recently (relative to other islands) been a factor.

- the analyses depend very much on the number of species under consideration, yet there is no mention of taxonomic uncertainty related to species delimitation. This should be explicitly addressed, at least in the text

- divergence times for the Malagasy lineages are implied to be important though it was difficult for me to determine the source for the inferred ages. This should be explicitly mentioned somewhere in the Methods

- the very interesting point that the vast majority of colonizations are within the bat clade –though perhaps not surprising – is nonetheless a key outcome of the study. This deserves mention in the Abstract, I would think

Reviewer #3 (Remarks to the Author):

The manuscript “The macroevolutionary impact of recent and imminent mammal extinctions on Madagascar” by Michielsen and colleagues represent an advance in our understanding of the impacts of human activities on the endemic mammalian fauna of Madagascar, it computes the times needed to recover these evolutionary loss and provide an estimate of the times that will be needed to recover this unique diversity if all species of mammals that are currently assessed (under IUCN red listing) in one of the threatened categories will go extinct in the near future. Madagascar host a unique and hiperdiverse flora and fauna which are both at high risk of extinction due to the strong pressures posed by the numerous societal challenges that this country is currently facing. Studies such as this one, can play a pivotal role in bringing the issue of the conservation of the unique biodiversity of Madagascar to the international attention in an effort to scale up current conservation efforts.

The manuscript is well written, methods are robust and the interpretation of the results is appropriate and the study has the potential to influence the adoption of this new metrics (ERT) in a conservation prioritization framework, as such I strongly recommend this manuscript for publication in Nature Communication

I have some minor comments to pass to the authors:

Intro line 49-51: I do not think this is less well known. Also maybe here you mean “since human expansion”?

Intro line 95: I think the debate regarding the date humans first established on the island is only marginally relevant to this study, as what matter is when human population expanded and started to have strong impacts on ecosystems. I think this intro should be a bit reworked to describe these two key aspects

Intro line 100: consider add some words here specifying that this (ca. 2500) is the timeframe during which you consider humans started to have an impact on Malagasy ecosystems... in your analyses

Intro line 101: how these uncertainties are expected to impact your priors and results?

Results line 133: consider add “and expansion” after “human arrival”

Figure 1: I think the figure will improve if 1) the column “threatened species IUCN 2021” in yellow on the right will be modified to represent the current assessment (2021) for all the species belonging to each independent colonisation event. Authors can use the same colour code as the one already used in the inset; 2) bat will be included in the inset; 3) in inset, Not evaluated species should be coloured differently from DD species

Results line 314: can you please add reference to table(s) and figure(s) for the 8.7 (6.9 - 20.7) Myr estimate?

Results line 317: can you please add reference to table(s) and figure(s) for the 26.2 (20.8 - 32) Myr estimate?

Results line 318: can you please add reference to table(s) and figure(s) for the 6.6 (5.5 - 7.8) Myr estimate?

Discussion line 369: consider substitute “settled” with “humans started to have an impact on Malagasy ecosystems”

Discussion line 382-390: thinking out loud here, but possibly this is can also be the result of the old age of Malagasy lineages (especially if considering the three main linages of non-volant mammals), which through the millions of years went through several environmental and geological changes, which could have resulted in an improved resilience to changes for the currently extant species.

M&M line 487: consider add “and expansion” after “colonisation”

M&M line 490 : is really " past few decades" that the authors meant? Please double check that what you want to say here is not "past few centuries".

Extended M&M line 800-801 please double check if one or more words are missing in this sentence

Extended M&M line 859: not sure I got this correctly, in case I did get correctly, why only adding the extinct species and not also the other 11 extant species (since from these you have even a better idea of their closely related species, as most of them are the results of recent systematics revisions)?

Extended M&M line 891: A non-endemic species can still be the result of diversification within Madagascar (aka it has its closest relative in a species endemic to Madagascar). Can you please confirm that none of the non-endemics has its closest relative in Madagascar? Might be worth to add this info to the Sup. Methods

Extended M&M line 1055: please double check that this is not Table S4 instead

Extended M&M line 1157: change to "extinct or"

Extended M&M line 1166: change "remain to be evaluated" to "are Not Evaluated or assessed as Data Deficient"

Figure S4 line 1255: add "in the next 10 years"

Figure S4 line 1257: ERTs?

Table S6 change "IUCN 2010" with "IUCN 2010 extinct to current"; "IUCN 2015" with "IUCN 2015 extinct to current"; "IUCN 2021" with "IUCN 2021 extinct to current"

Table S7 change "IUCN 2010" with "IUCN 2010 extinct to current"; "IUCN 2015" with "IUCN 2015 extinct to current"; "IUCN 2021" with "IUCN 2021 extinct to current"

I do not need to remain anonymous,

Sincerely,

Angelica Crottini

Reviewer comments

Reviewer #1

In their manuscript the authors use an island biogeography model to evaluate the impact of previous and future extinctions of non-volant mammals and bats in Madagascar. I think that the article is well-written, well-structured and, although I am not an expert in macroevolutionary data and models, includes a thorough compilation of phylogenetic data for Malagasy mammals. In the body of the paper, the methods are clear and the results and conclusions follow logically. The paper contributes to understanding the effects of losing species (diversity) which is important in terms of management and policy considerations.

We thank the reviewer for this positive assessment.

However, there are three aspects of the paper which could potentially be improved and/or on which I believe the authors could provide more detail. First, I understand the reasoning why the authors have used constant CES rates through time but I question the usability of constant rates based on historical data for future predictions. Would it also be possible to use non-constant rates over time? For example, could the authors use rates of more recent times to predict the ERT when all threatened species would go extinct? Recent colonization rates might for example be more suitable for future predictions than rates based on the entire time span, as humans may have (had) a positive influence here. This would, in my opinion, reflect reality better.

We thank the reviewer for this suggestion. We have performed a new analysis where we allow for the rates of colonisation to shift (increase or decrease) at a certain point in time. We fitted 2 new models: M31, where bats and mammals share the same rates but underwent a shift in colonisation rate at some point in time; and M32, where bats and non-volant mammals have different rates, and each underwent a shift in colonisation rate at some point in time. We find that the shift models are not preferred using AIC or BIC (Table S3), so the preferred model remains a constant rates model (M26). We cannot rule out that a multi-shift model, or a model with other forms of temporal variation in rates would fit the data better, but we currently do not have a likelihood estimation procedure for those models and this would merit a different methodological study on its own (for example, see Hauffe et al 2020 *Journal of Biogeography* for the analyses where we developed and extensively tested the performance of the single-shift model).

Regarding the reviewer's suggestion to use recent rates to calculate the ERT, such an approach differs from how we conceptualised the ERT. If we estimate ERT based on recent human-influenced rates (e.g. higher colonisation due to new human-altered habitats emerging), the ERT would not reflect the natural biogeographical rates in the absence of humans, which is what we aim to model. In addition, the new analyses mentioned above do not find evidence for a change in rates in recent times, and recent changes in rates driven by human influences would not be reflected in our data, because we purposely excluded species that were introduced by humans (directly or

indirectly). We have added the new analyses and now discuss these points in the main text:

“We did not find evidence for temporal rate variation (models M31 and M32, Table S3) and we thus estimated ERTs assuming the average rates across the entire history of the island in the absence of humans. Models with multiple temporal shifts in rates or with other forms of temporal variation are yet to be developed, but we nevertheless favour our use of average natural constant rates (without anthropogenic interference) as this allows for standardisation when comparing with the ERT for islands in different biogeographical contexts.”

Second, related to the first point, I was not sure after the reading the MS in what way colonization exactly contributes to species diversity in DAISIE. Is it correct that colonization could directly lead to higher diversity values when the particular species was not yet occurring (or has gone extinct) on Madagascar? I am then wondering whether the approach should not focus on endemic species only and thus indirect contributions of colonization. In other words, ERT (to me) seems to be more valuable to investigate the global loss of species instead of looking at local diversity. So should you then take colonization even into account apart from the routes it may provide for cladogenesis and anagenesis?

This is a great point. We now include a new analysis where we estimate the ERTs for reaching the number of endemic species. We include the analyses but we prefer to focus on the total diversity of Madagascar to simplify the message of the paper. We now mention in these analyses in the Results and Table S9:

“The ERTs we estimate focus on the local Malagasy diversity, not on global diversity. Global diversity will only increase when a new endemic species evolves on Madagascar. For example, colonization of Madagascar by a mainland species can lead to higher diversity values on the island if the species is not already present there, but global diversity would not change. This scenario will be particularly common when colonisation rates are high and speciation rates are low. However, in the estimated parameters of our model for non-volant mammals, the rates of speciation (both anagenesis and cladogenesis) are much higher than rates of colonization, and thus a return to a target “global” diversity (e.g. return to pre-human number of endemic species) is achieved in a similar time frame as a target local diversity (Table S9). In the case of bats, a target “global” diversity is achieved more rapidly than the target local diversity, because both colonisation and speciation rates are high and the number of endemic species to recover is low (Table S9).”

Finally, after reading the MS, I would have a few remaining questions if I was a conservationist regarding the applicability of the results. I think it is very important to know what would be the consequences of losing insular species and (as the authors describe) the ERT can be used to compare different islands to see where effects are most detrimental. But now that we know that losing species on Madagascar will take a long time to recover, what should conservationists do next? Which species and areas on Madagascar should be protected/conserved? I do not expect the authors to answer these questions in the MS but I do feel that these important aspects should be discussed. What in my opinion would help in this regard, is to investigate the (potential)

relationship between clade/species characteristics and ERT. In lines 974 - 976 the authors state that a trait-dependent diversification model for insular communities not yet exists, but could the authors not run simulations with different species diversity (and thus different missing species) to find out what species characteristics/lineages prove to be more difficult to replace (similar to the approach described in lines 1120 - 1123)? This would greatly increase the significance of the paper by helping to understand which species are especially difficult to 'replace', and, in turn, which should receive conservation focus (also potentially outside of Madagascar).

We have now run an analysis where we estimate the ERT for each of the major clades, by measuring how long it would take to recover diversity back to contemporary diversity if the currently threatened species (IUCN 2021) in each of those groups go extinct. This analysis must be viewed with caution because the model assumes equal rates among clades (except bats versus non-volant mammals). As the simulations are run on the entire island community, groups that have the same number of threatened species will necessarily have the same ERT, which is likely not the case in reality.

We have added the following text to the Results:

“We also ran ERT analyses separately for each of the major groups (bats, carnivorans, lemurs, rodents, and tenrecs), by measuring how long it would take to recover diversity back to contemporary levels if currently threatened species (IUCN 2021) in those groups go extinct. Because rates are constant between lineages in our model (except between bats and non-volant mammals), groups with similar numbers of threatened species will have similar ERTs. Using this approach, we find that the ERT is longest for the lemurs (recovering 104 species, 17.47 (13.81-21.3) Myr), followed by the bats (5 species, 2.91 (2.32-3.6) Myr), the Nesomyinae rodents (7 species, 0.8 (0.63-0.99) Myr) and lastly the carnivorans and tenrecs (6 species to recover each, 0.69 (0.54-0.84) Myr).”

And to the Discussion:

“Given the taxonomic and trait diversity of mammals on Madagascar, the size of the island, high levels of habitat heterogeneity, and the fact that all of the native mammal species are endemic, many with very limited geographic distributions, future studies could aim to identify and recommend particular clades and regions that should be the focus of heightened conservation activities to minimise an incipient wave of extinctions. Based on our results alone we do not recommend prioritising conservation management of one clade over the others, but our finding that ERTs are highest for lemurs and bats reinforces the urgent need to protect lemurs given the amount of time it would take to recover their diversity (> 13 Myr), and reveals that Malagasy bats are also a group of particular concern, at least from an evolutionary perspective (> 2 Myr).”

Minor points:

-The authors have shown in their MS that the species lost is not a good surrogate for ERT and that the link between the two is non-linear. The assumption made by the authors for future predictions now is that all threatened species will go extinct, no matter the difference in IUCN categories, whereas it is likely that CR species will go extinct earlier compared to VU species. How would this affect the results?

We have now added a new analysis where we assume that only CR species go extinct. In the case of bats there are no CR species currently listed, so the results are unchanged. In the case of non-volant mammals, if only CR species go extinct, ERTs decline substantially (see new Table S8, scenario B).

-Lines 64 - 66: to me the interrogative sentences read a bit strangely so I would suggest to rewrite these.

We have changed these to declarative sentences.

-Lines 98 - 99: I would suggest to remove 'in a conservative manner' here. Based on the previous sentence, I would argue that the authors are not that conservative here as they state that humans "largely" disappeared for 8000 years.

We agree and have rewritten this sentence.

-Lines 118 - 120: suggest to rewrite as an increase in threatened species is not directly related to ERT or to the number of species lost (only with the assumption made by the authors that all threatened species will go extinct in the near future which has not been stated in the MS yet).

We have rephrased to: *"Whether this recent increase in threatened species would have a disproportionate impact on estimates of ERT if these species eventually go extinct will depend on the extent to which the number of species lost is a good surrogate for ERT."*

-Lines 757 - 758: I would appreciate if the authors can include the timing of extinction in Supplementary Data S1. This makes it more easy as a reader to follow the assumption made in the MS.

Great suggestion. We have added a new column with the last ¹⁴C date of appearance in the fossil record.

-Line 922: change colonisation to colonization.

Done.

-Table S3: please also report the AIC and BIC values here which were used to select the best model.

Done.

-Table S6: based on the description of the data sets and the parameters therein, I would assume that: 1) the larger the mainland pool the lower the ER, and 2) CS2 would lead to a lower ERT compared to CS1. This is not (always) the case. Can you explain why? Similarly, why would the ERT for non-volant mammals be higher in D4 compared to D1? The opposite is true for D5 vs D2 and D6 vs D3.

We thank the reviewer for looking into this in detail. In Table S6 for D2-D13 we showed the results from the maximum-clade credibility (MCC) trees, but for D1 we had decided to show the mean values from the posterior distribution (which is what we report elsewhere in the text). In hindsight, we realise this can be confusing, so in the revised version we now show in Table S6 only results from the MCC trees (also for D1). We nevertheless caution against making guesses of ERT based on the parameters, because the differences in parameters are very subtle and small changes can lead to significant differences in ERT. For example, a higher mainland pool leads to a lower colonisation rate, which should increase the ERT. However, changes in the mainland pool can also lead to very small changes in the diversification rate (cladogenesis minus extinction), which has a much higher impact on the ERT than colonisation rate because colonisation is very rare. We have now added sentences to the Supplementary Methods section “Evolutionary return times” to highlight this.

-Lines 1034 - 1035: Related to my previous point, you state here that the colonization rate decreases with mainland pool size. Why is this? Simply because the unit for colonization rate that you used is per mainland species? Then it would be good to specify. Or is it because of homogenization?

This is because the colonization rate unit is per mainland species. We now added this in the text.

-Line 1157: change 'extinct or' to 'extinct or'.

Done

-Figure 1: I would suggest to shade the filled symbols on top of the dashed lines to highlight that these are estimates/random.

Done.

-Line 240: bats follow an equilibrium model towards 57 species. How does that relate to a carrying capacity of an infinite number of species in the model? It is surprising to me that this model (with setting K to infinity) then performs better than when the carrying capacity was an emerging property for bats. Can you explain?

The model with a carrying capacity (diversity-dependent equilibrium) is quite different from the sink equilibrium model. In the former, species compete for limited niches, there is an upper bound to diversity set by the carrying capacity (K), and rates of cladogenesis and colonisation decline with increasing diversity on the island. In the latter, the equilibrium is not driven by diversity-dependence, and is driven by the fact that the rate of extinction is high. We have now added the following sentence:

"The sink equilibrium detected in bats differs from a diversity-dependent equilibrium model. Under diversity dependence there is an upper bound to species diversity on the island driven by competition for limited niches and set by a diversity carrying capacity (K), but this was rejected for bats. In the sink scenario, the equilibrium is not driven by diversity dependence in rates, but instead controlled by the high rate of extinction."

-Figure 2: from the figure I conclude that the diversity goals (e.g. pre-human diversity) are set separately for bats and non-volant mammals, right? In other words, when the pre-human diversity for bats of 44 is reached, the additional/new bat species (from the found limit of 57 species) do not contribute to reaching the goal, right? I would do specifically state this in the main text.

Indeed that is the case. We have now added the following sentence “*Note that pre-human and contemporary diversities differ between non-volant species and bats (Fig. 3, S3 and Table S5), and that once these “target” diversities have been reached, diversity can continue to increase (up to the equilibrium diversity in case there is one, as is the case in bats).*”

NB: are figures 2C and 2D needed?

We think the four panels work quite well together as a visual Abstract. Alternatively, we could extract panels C+D and create a new main text figure. We will ask the editorial team about this.

-Line 267: suggest to change to 'only 91 species of mammals out of a total of 219 mammals (current diversity) would remain on the island'.

Done.

-Figure 2 and 3: I would suggest to use only one of these figures in the main text as the information they hold overlaps. Note that the authors always also report to both of these figures when discussing the results.

While we agree that there is some redundancy between the two figures, they show the ERTs from very different perspectives. We really like both figures, and would prefer to maintain them in the main text.

Reviewer #2

This manuscript offers a novel perspective on the extinction crisis in Madagascar by using quantitative approaches that integrate speciation times, inferred dispersal events, geological context, and human impacts. Notably, it also focuses on bats which have been under appreciated in conservation analysis in Madagascar and elsewhere. Though I am not familiar with the primary analytical method (DAISIE), the assumptions seem sound.

I believe that this will be an important contribution to the extinction/conservation literature in that it makes the evolutionary costs of human-mediated extinction abundantly clear.

We thank the reviewer for the positive assessment.

I have a few minor comments for improvement:

- the sentence in the introduction (lines 51 - 55) relating to human impacts is confusing, I think due to the word "protracted." If I understand the meaning of the sentence correctly, it seems like a word like "delayed" would be better suited to the point that biodiversity persists given that human impacts have only recently (relative to other islands) been a factor.

We have changed this to "delayed".

- the analyses depend very much on the number of species under consideration, yet there is no mention of taxonomic uncertainty related to species delimitation. This should be explicitly addressed, at least in the text

We thank the reviewer for this suggestion. In our analysis "Impact of increased knowledge on the ERT", we had already directly dealt with taxonomic uncertainty by assuming the number of species may increase if species are split, but we had only mentioned this in the supplementary. We now mention this also in the main text *"Previously undescribed extinct and extant species of mammals are likely to be discovered on Madagascar in the near future, and taxonomic revisions may lead to species splits, resulting in additional threatened species. We therefore also estimated the impact this may have in our ERT calculations..."*.

- divergence times for the Malagasy lineages are implied to be important though it was difficult for me to determine the source for the inferred ages. This should be explicitly mentioned somewhere in the Methods

The source of divergence times is the phylogeny of Upham et al. 2019. We now make this clearer in the main text Methods. Please note that the source of the divergence times and our approach to extract them from the phylogenetic trees are explained in detail in the Supplementary Methods.

- the very interesting point that the vast majority of colonizations are within the bat clade –though perhaps not surprising – is nonetheless a key outcome of the study. This deserves mention in the Abstract, I would think

Because we have reached the word limit in the Abstract, we now highlight this point in the Results with a new sentence: *"Under both scenarios, the number of colonisations of bats greatly exceeds that of non-volant mammals."*

Reviewer #3

The manuscript “The macroevolutionary impact of recent and imminent mammal extinctions on Madagascar” by Michielsen and colleagues represent an advance in our understanding of the impacts of human activities on the endemic mammalian fauna of Madagascar, it computes the times needed to recover these evolutionary loss and provide and estimate of the times that will be needed to recover this unique diversity if all species of mammals that are currently assessed (under IUCN red listing) in one of the threatened categories will go extinct in the near future. Madagascar host a unique and hiperdiverse flora and fauna which are both at high risk of extinction due to the strong pressures posed by the numerous societal challenges that this country is currently facing. Studies such as this one, can play a pivotal role in bringing the issue of the conservation of the unique biodiversity of Madagascar to the international attention in an effort two scale up current conservation efforts.

The manuscript is well written, methods are robust and the interpretation of the results is appropriate and the study has the potential to influence the adoption of this new metrics (ERT) in a conservation prioritization framework, as such I strongly recommend this manuscript for publication in Nature Communication

We thank the reviewer for the positive assessment and constructive comments.

I have some minor comments to pass to the authors:

Intro line 49-51: I do not think this is less well known.

We have rephrased this sentence “*Like most islands, Madagascar underwent substantial levels of extinction...*”.

Also maybe here you mean “since human expansion”?

We have added “since human arrival and population expansion”.

Intro line 95: I think the debate regarding the date humans first established on the island is only marginally relevant to this study, as what matter is when human population expanded and started to have strong impacts on ecosystems. I think this intro should be a bit reworked to describe these two key aspects

We now refer explicitly to population expansion and potential impact on ecosystems throughout the text.

Intro line 100: consider add some words here specifying that this (ca. 2500) is the timeframe during which you consider humans started to have an impact on Malagasy ecosystems... in your analyses

We have rewritten this sentence: “*Thus we consider the period of established and continuous human habitation to be the time frame encompassing ~2500 years BP onward, when the human population expanded and started to have a strong impact on the island’s ecosystems.*”

Intro line 101: how these uncertainties are expected to impact your priors and results?

We deal with this point in detail in our analyses of low versus high human impact. We prefer to refer to these at a later stage in the manuscript.

Results line 133: consider add “and expansion” after “human arrival”

Done.

Figure 1: I think the figure will improve if

1) the column “threatened species IUCN 2021” in yellow on the right will be modified to represent the current assessment (2021) for all the species belonging to each independent colonisation event. Authors can use the same colour code as the one already used in the inset;

Our preference is to refer to Supplementary Data 1 for this information, as this would require fitting eight categories into the middle column, which would be too much to condense in a figure.

2) bat will be included in the inset;

3) in inset, Not evaluated species should be coloured differently from DD species

We have revised Figure 1 to incorporate these 2 great suggestions.

Results line 314: can you please add reference to table(s) and figure(s) for the 8.7 (6.9 - 20.7) Myr estimate?

Results line 317: can you please add reference to table(s) and figure(s) for the 26.2 (20.8 - 32) Myr estimate?

Results line 318: can you please add reference to table(s) and figure(s) for the 6.6 (5.5 - 7.8) Myr estimate?

For all the above: we had previously reported these values only in the text, but we agree a table is a good idea, so we have now added a new Table S8 showing these results.

Discussion line 369: consider substitute “settled” with “humans started to have an impact on Malagasy ecosystems”

Done.

Discussion line 382-390: thinking out loud here, but possibly this is can also be the result of the old age of Malagasy lineages (especially if considering the three main lineages of non-volant mammals), which through the millions of years went through several environmental and geological changes, which could have resulted in an improved resilience to changes for the currently extant species.

We thank the reviewer for this suggestion. We prefer to not mention this as it would require us to obtain the ages of lineages on other islands for comparison. Some very old lineages on other islands (e.g. the moa of New Zealand and the sloths of the West

Indies) did suffer substantial extinctions after human arrival, so this relationship is not straightforward.

M&M line 487: consider add “and expansion” after “colonisation”

Done.

M&M line 490 : is really “ past few decades” that the authors meant? Please double check that what you want to say here is not “past few centuries”.

We indeed meant the past few decades, to highlight the intensification of impact that has occurred.

Extended M&M line 800-801 please double check if one or more words are missing in this sentence

We have rephrased for clarity.

Extended M&M line 859: not sure I got this correctly, in case I did got correctly, why only adding the extinct species and not also the other 11 extant species (since from these you have even a better idea of their closely related species, as most of them are the results of recent systematics revisions)?

The 11 extant species were also added, using the same methodology as used for the extinct species. We have now added to the text the number of missing extinct species (23), which we hope will help make this clearer.

Extended M&M line 891: A non-endemic species can still be the result of diversification within Madagascar (aka it has its closest relative in a species endemic to Madagascar). Can you please confirm that none of the non-endemics has its closest relative in Madagascar? Might be worth to add this info to the Sup. Methods

Indeed, three of the non-endemic species resulted from diversification within Madagascar and then colonised other regions, so their closest relatives and origin are on Madagascar. These are three species of bat in the genus *Miniopterus*. We have now added this information to the text. In our analyses and figures these three species are treated correctly and contribute to cladogenesis rates on Madagascar.

Extended M&M line 1055: please double check that this is not Table S4 instead

We thank the reviewer for noticing this, indeed both tables are useful in this case.

Extended M&M line 1157: change to “extinct or”

Done.

Extended M&M line 1166: change ”remain to be evaluated” to “are Not Evaluated or assessed as Data Deficient”

Done.

Figure S4 line 1255: add “in the next 10 years”

Done.

Figure S4 line 1257: ERTs?

Done.

Table S6 change “IUCN 2010” with “IUCN 2010 extinct to current”; “IUCN 2015” with “IUCN 2015 extinct to current”; “IUCN 2021” with “IUCN 2021 extinct to current”

Done.

Table S7 change “IUCN 2010” with “IUCN 2010 extinct to current”; “IUCN 2015” with “IUCN 2015 extinct to current”; “IUCN 2021” with “IUCN 2021 extinct to current”

Done.

I do not need to remain anonymous,

Sincerely,
Angelica Crottini

REVIEWERS' COMMENTS

Reviewer #1 (Remarks to the Author):

The authors have satisfactorily addressed all of my comments (and those of the other reviewers). The responses to the comments are clear and the changes made to the text improved the MS. I therefore support and recommend publication of this study.

My only final doubt is whether the added analysis on the estimation of the ERT for each of the major clades adds much to the MS. When the rates are constant between the clades, the differences in ERT will only be because of a difference in number of threatened species, right? I know that the authors have added this analysis because of one of my comments, but for me it would also suffice if only lines 548 - 553 (on future studies) remained in the MS. I leave this choice up to the authors and/or editorial team.

Reviewer #2 (Remarks to the Author):

The authors have done an excellent job of responding to my and the other reviewers' suggestions for improvement. I recommend prompt acceptance.

Reviewer #3 (Remarks to the Author):

[Editor's note: the reviewer had no further comments for the authors and recommended acceptance]

REVIEWERS' COMMENTS

Reviewer #1 (Remarks to the Author):

The authors have satisfactorily addressed all of my comments (and those of the other reviewers). The responses to the comments are clear and the changes made to the text improved the MS. I therefore support and recommend publication of this study.

We thank the reviewer for their positive comments.

My only final doubt is whether the added analysis on the estimation of the ERT for each of the major clades adds much to the MS. When the rates are constant between the clades, the differences in ERT will only be because of a difference in number of threatened species, right? I know that the authors have added this analysis because of one of my comments, but for me it would also suffice if only lines 548 - 553 (on future studies) remained in the MS. I leave this choice up to the authors and/or editorial team.

Following the reviewer's suggestion we have now removed the new analysis and left the original sentences instead.

Reviewer #2 (Remarks to the Author):

The authors have done an excellent job of responding to my and the other reviewers' suggestions for improvement. I recommend prompt acceptance.

We thank the reviewer for their positive recommendation.

Reviewer #3 (Remarks to the Author):

[Editor's note: the reviewer had no further comments for the authors and recommended acceptance]

We thank the reviewer for their positive recommendation.